

# A numerical study of tsunami wave run-up and impact on coastal cliffs using a CIP-based model

Xizeng Zhao[1], Yong Chen[1], Zhenhua Huang[2], Yangyang Gao[1]

[1] Ocean College, Zhejiang University, Zhoushan Zhejiang 316021, China
[2] Department of Ocean and Resources Engineering, School of Ocean and Earth Science and Technology, University of Hawaii at Manoa, USA

*Correspondence to:* Xizeng Zhao (xizengzhao@zju.edu.cn)

**Abstract.** There is a general lack of the understanding of tsunami wave interacting with complex geographies, especially the process of inundation. Numerical simulations are performed to understand the effects of several factors on tsunami wave impact and run-up in the presence of submarine gentle slopes and coastal cliffs, using an in-house code, named a Constrained Interpolation Profile (CIP)-based model in Zhejiang University (CIP-ZJU). The model employs a high-order finite difference method, the CIP method as the flow solver, utilizes a VOF-type method, the Tangent of hyperbola for interface capturing/Slope weighting (THINC/SW) scheme to capture the free surface, and treats the solid boundary by an immersed boundary method. A series of incident waves are arranged to interact with varying coastal geographies. Numerical results are compared with experimental data and good agreement is obtained. The influences of submarine gentle slope, coastal cliff and incident wave height are discussed. It is found that the rule of tsunami amplification factor varying with incident wave is affected by angle of cliff slope, and there is a critical angle about 45°. The run-up on a toe-erosion cliff is smaller than that on a normal cliff. The run-up is also related to the length of submarine gentle slope with a critical about 2.292m in the present study. The impact pressure on the cliff is extremely large and concentrated, and the backflow effect is nonnegligible. Results of our work are in high precision and helpful in inversing tsunami source and forecasting disaster.

## 1. Introduction

Tsunami one of the coastal hazards in the world, can be caused by earthquake, volcanic eruption and submarine landslide. The 2004 tsunami in Southeast Asia is one of the most destructive tsunami events in human history. There were over one hundred thousand victims in 11 countries during the tsunami period (Liu et al., 2005). In the recent Great East Japan Earthquake and Tsunami in 2001, over 24 thousand people were killed or missing and 300 thousand building damages occurred (Mimura et al., 2011). More serious nuclear disaster at the Fukushima Nuclear Power Plants No.1 was caused by the powerful run-up and destructive force of the tsunami wave. All these events and lessons from the previous tsunami wave disasters indicate that the actual tsunami wave run-up and the corresponding destructive forces were underestimated (Dao et al., 2013).



Investigation of tsunami wave transformation in the near shore area is a feasible approach to learn the action mechanism and to inverse the tsunami source. Post-disaster studies were mostly done through field observations and numerical simulations. The sediments in the near shore area are frequently regarded as traces of a tsunami. Monecke et al. (2008) analyzed sand sheet deposited by the 2004 tsunami and extended tsunami history 1,000 years into Aceh past, pointing out the recurrence

frequency of a damage-causing tsunamis. However, due to the strong backflow of tsunami, sediment can be brought back to seaward. Hence, there may be an underestimation of the run-up using sediment information to study paleotsunamis. Dawson (1994) pointed out that the upper limit of sediment deposition lay well below the upper limit of wave run-up which was marked by a well-defined zone of stripped vegetation and soil. Goto et al. (2011) found that previous estimates of paleotsunamis have probably underestimated by considering their newly acquired data on the 2011 Japanese tsunami event.

However, limited by the complexity in field observations, numerical simulation is another effective approach to investigate tsunami. The computational domain should include a large area in which tsunami generates, propagates and inundates (Sim et al., 2005). Therefore, shallow water equations (SWE) were popularly used due to the high efficiency. However, it is incompatible that when there is interaction between wave and complex geography, SWE is unable to capture flow structures in detail and often underestimates the result (Liu et al., 1991). An amendment is needful to improve the result

of SWE with the data from physical experiment or field observation.

       Since the available time history data of tsunami waveform and flow field in the near-shore area is scarce, there is a general lack of understanding of tsunami interacting with complex geography. A more accurate method is required to reproduce the process of tsunami evolution in the coastal area. Due to the development of supercomputing technology and precise numerical algorithm, computational fluid dynamics (CFD) with viscous flow theory and fluid-solid coupling mechanics are

capable of dealing with the complex flow problems when geographies exist. It is a significant research project to deal with the free surface in CFD. Hirt et al. (1981) put forward a mass conservation method named Volume of Fluid (VOF). Based on the principle of VOF, several improved methods were developed: PLIC (Youngs, 1982), THINC (Xiao et al., 2005), WLIC (Yokoi, 2007) and THINC/SW (Xiao et al., 2011). When coastal geographies are included, special handling is required. Peskin (1973) proposed an immersed boundary method (IBM) to treat the blood flow patterns of human heart, and was later

introduced to simulate the interactions between solid objects and incompressible fluid flows (Ha et al., 2014; Lin et al., 2015).

       CFD is more convenient comparing with experimental data or field observation. The most valuable achievement is that it can give time history data of waveform, pressure and flow velocity field to understand the evolution and impact mechanism of a tsunami in near-shore areas. When efficient numerical algorithms adopted, a CFD-based numerical simulation with a

suitable scale can be applied to study the tsunami inundation in coastal areas. The purpose of present work is to understand the destructive nature of a tsunami, helping inverse the generation mechanisms and provide reference for the tsunami forecasting and post-disaster treatment.

       Coastal cliffs are one of the common coastal landforms, representing approximately 75% of the world's coastline (Rosser et al., 2005), such as the coast of Banda Aceh in Indonesia and the steep slope at San Martin (António et al., 1993). The



existence of cliff can not only influence the impact and run-up of a tsunami wave but also the erosion-deposition. Different layers provide variations in resistance to erosion (Stephenson et al., 2011). Particularly, some coastal cliffs consisting of soft rocks are eroded at the toe (Yasuhara et al., 2002), which makes it easier to be destroyed. It is indispensable to understand the function of coastal cliffs and submerged gentle slopes during the tsunami wave approaches near shore. In this study,

tsunami wave run-up and impact on coastal cliffs is simulated using an in-house code, named the CIP-ZJU model. Considerable attention is paid to the influence of different coastal topographies on the tsunami inundation in near shore areas. Steep cliffs on the beach and submerged gentle slopes are considered. The submerged gentle slope such as the continental shelf, affects the waveform evolution and wave celerity before tsunami waves reach the shoreline. Tsunami amplification factor, relative wave height, run-up on the cliff and impact pressure will be analyzed in this work.

In this paper, Section 2 describes the governing equations and the numerical methods; Section 3 presents the initial condition and model validation; Dimensionless analysis is then used to examine the effect of front slope length, depth ratio and cliff angles on the run-up, and impact pressure; finally, the conclusion is made in Section 4.

## 2 Numerical models

### 2.1 Governing equations

Our model is established in a two-dimension Cartesian coordinate system, based on viscous fluid theory with incompressible hypothesis. The governing equations are continuity equation and Navier-Stokes equations written as follows

$$\nabla \cdot \boldsymbol{u} = 0 \tag{1}$$

$$\frac{\partial \boldsymbol{u}}{\partial t} + (\boldsymbol{u} \cdot \nabla)\boldsymbol{u} = -\frac{1}{\rho}\nabla p + \frac{\mu}{\rho}\nabla^2\boldsymbol{u} + f \tag{2}$$

where $\boldsymbol{u}$, $t$, $\rho$, $p$, $\mu$ and $f$ are the velocity, time, fluid density, hydrodynamic pressure, dynamic viscosity and momentum

forcing components, respectively.

Multiphase flow theory is employed to solve the problem of solid-liquid-gas interaction. A volume function $\phi_m$ is defined to describe the percentage of each phase in a mesh

$$\frac{\partial \phi_m}{\partial t} + \boldsymbol{u} \cdot \nabla \phi_m = 0 \tag{3}$$

where $m = 1, 2, 3$, indicating liquid, gas, solid respectively, and $\phi_1 + \phi_2 + \phi_3 = 1$.

Physical property, such as the density and viscosity in a mesh can be calculated by:

$$\lambda = \sum_{m=1}^{3} \phi_m \lambda_m \tag{4}$$



## 2.2 Numerical methods

A fractional step approach is applied to solve the time integration of the governing equations (1) and (2). The first step is to calculate the advection term, neglecting the diffusion term and pressure term, as equation (5) shows.

$$\frac{\partial \boldsymbol{u}}{\partial t} + (\boldsymbol{u} \cdot \nabla)\boldsymbol{u} = 0 \tag{5}$$

A CIP (Constrained Interpolation Profile) method is employed to solve equation (5). The CIP method was first introduced by Takewaki et al. (1985) as an efficient method to solve the hyperbolic partial differential equation. The basic principle of CIP is an interpolation in a grid by a cubic polynomial with the value and differential on the grid node. The advantage of CIP is that it can provide a third order interpolation function in a single grid, which makes it a compact high order scheme.

The second step is to solve the diffusion term by a central difference scheme

$$\frac{\boldsymbol{u}^{**} - \boldsymbol{u}^{*}}{\Delta t} = \frac{\mu}{\rho} \nabla^2 \boldsymbol{u} + \vec{F} \tag{6}$$

where $\boldsymbol{u}^{*}$ is the solution of equation (5), and $\boldsymbol{u}^{**}$ is the solution to calculate in this step.

The final step is the coupling of the pressure and velocity by considering equation (1):

$$\nabla \cdot (\frac{1}{\rho} \nabla p^{n+1}) = \frac{1}{\Delta t} \nabla \cdot \boldsymbol{u}^{**} \tag{7}$$

$$\boldsymbol{u}^{n+1} = \boldsymbol{u}^{**} - \frac{\Delta t}{\rho} \nabla p^{n+1} \tag{8}$$

Equation (7) is solved by a successive over relaxation (SOR) method. More details can be found in our previous works (Zhao et al., 2016a and 2016b).

The free surface is captured by a tangent of hyperbola for interface capturing with slope weighting (THINC/SW) scheme, which is based on the principle of VOF method. The THINC/SW was put forward by Xiao et al. (2011), using a hyperbolic tangent function with adjustable parameter to interpolate. The solid boundary is treated by an immersed boundary method (IBM) (Peskin et al., 1973).

## 3 Numerical results

### 3.1 Numerical wave tank

In this section, 2D numerical wave tanks including incident waves and geographies are introduced. Simulation cases are divided into two categories according to different parameters and purposes.

The first simulations are completed in 1# tank, as shown in Fig. 1. This wave tank is 10.0 m in length and 1.0 m in height. Four slopes compose the topography-profile, representing continental slope, continental shelf, beach and cliff, respectively. The still water depth in front of the topography-profile is fixed at 0.35 m, so that the still-water shoreline is located at the starting point of the beach. This point is regarded as the original point of this tank to determine other positions mentioned in



this paper. 1# tank is used for verifying the accuracy of our model and investigating the effect of the angle of cliff slope and incident wave height on the tsunami amplification factor. The angles of slopes are $\tan\theta_1$= 25/17, $\tan\theta_2$= 1/15, $\tan\theta_3$= 1/30. Five angles of cliff slope are tested: $\theta_4$ = 14°, 21.67°, 39.33°, 49° and 79°. And four incident wave heights are considered in this tank: $H$ = 0.025 m, 0.035 m, 0.045 m and 0.055 m. Solitary waves are used as initial condition in the numerical

modelling of tsunami. S1~3 in Fig. 1 are three measuring gauges of water elevation for the comparison of numerical result and experimental data, located at $x$= -7.67 m, -0.87 m and 0.11 m, respectively. Besides, six measuring gauges of water level elevation in front of the cliff are employed to analyze the tsunami amplification factor, fixed at $x$ = 0.0 m, 0.06 m, 0.11 m, 0.13 m, 0.16 m and 0.21 m, respectively (not draw in Fig. 1).

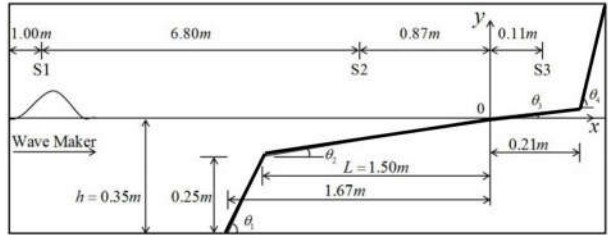

Fig. 1 Schematic diagram of 1# tank

The second simulations utilize the 2# tank, similar to the 1# tank except slight difference, as shown in Fig. 2 (a). In this tank, the still-water shoreline also lies on the starting point of the beach, which is the original point of this tank. The angles of slopes are $\tan\theta_1$ = 1.38, $\tan\theta_2$ = 0.08, $\tan\theta_3$ = 0.02. Four lengths of the submarine gentle slope (standing for continental shelf) are set: $L$ = 0.764 m, 1.528 m, 2.292 m and 3.056 m. Three incident wave heights are performed, $H$=0.04m, 0.05m,

0.06m. Two kinds of cliff: normal cliff $\theta_4$ = 80.02°, and toe-eroded cliff $\theta_4$ = 91.91° are considered. Six measuring points of water level elevation are employed to record the waveform evolution, located at $x$= -0.87 m, 0.0 m, 0.1 m, 0.2 m, 0.3 m, 0.4 m, respectively. Five pressure sensors are located near the toe of the cliff, as shown in Fig. 2 (b).

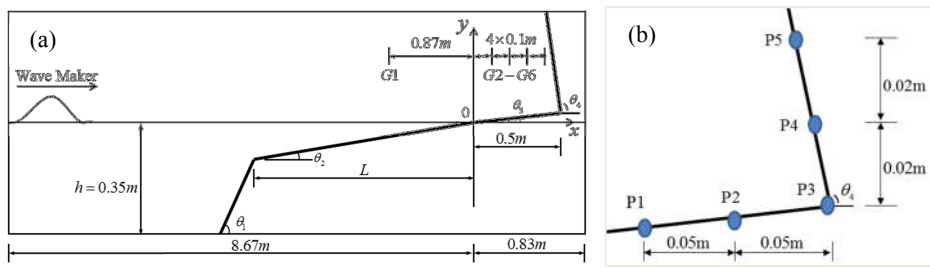

Fig. 2 Schematic diagram of 2# tank

In this work, considerable attention will be paid to 2# tank. It is necessary to number the simulated cases in 2# tank to avoid confusion, as shown in Table 1.



Table 1 Summary of basic parameters calculated in 2# tank

| Case | $\theta_4$=80.02° $H$=0.04 m | $\theta_4$=80.02° $H$=0.05 m | $\theta_4$=80.02° $H$=0.06 m | $\theta_4$=91.91° $H$=0.04 m | $\theta_4$=91.91° $H$=0.05 m | $\theta_4$=91.91° $H$=0.06 m |
|---|---|---|---|---|---|---|
| $L$=0.764 m | 1 | 5 | 9 | 13 | 17 | 21 |
| $L$=1.528 m | 2 | 6 | 10 | 14 | 18 | 22 |
| $L$=2.292 m | 3 | 7 | 11 | 15 | 19 | 23 |
| $L$=3.056 m | 4 | 8 | 12 | 16 | 20 | 24 |

### 3.2 Model validation

To verify the accuracy of our model, numerical result from one of the cases in 1# tank is compared with available experimental data (Sim et al. 2015). The incident wave height and the cliff slope angle of this case are $H$ =0.055m and $\theta_4$ = 79°, respectively. A variable grid is used for the computation, in which the grid points are concentrated near the free surface and the topography. Three non-uniform grids are used to perform a grid refinement test. The grid number and the minimum grid size are shown in Table 2. .

Table 2 Parameter of three sets of grids (Unit: m)

| | Grid number in $x$ direction | Grid number in $y$ direction | Minimum grid size in $x$ direction | Minimum grid size in $y$ direction |
|---|---|---|---|---|
| Coarse-grid | 826 | 220 | 0.008 | 0.0022 |
| Middle-grid | 970 | 320 | 0.005 | 0.0015 |
| Fine-grid | 1228 | 468 | 0.003 | 0.0008 |

Fig. 3 concerns the predicted time series of water elevations at different locations S1~3 and the physical measurements (Sim et al., 2015) are also presented for comparison. Fig. 3 (a) illustrates the comparison results at S1. It can be observed that wave has not reached the topography, so that the waveform has not transformed grossly and is similar to the original waveform. Good general agreement is found for all computations. The relative wave height at S1 is 1.0, which reveals the accuracy of the target incident wave. Fig. 3 (b) shows the results at S2 $x$ = -0.87 m. This gauge point is in the area of the submarine gentle slope, and shoaling happens when wave propagates here. It can be seen from Fig. 3 (b) that the wave front face becomes steep and the back face becomes gentle, which means wave asymmetry appears. Results of three grids are in good agreement with experimental data. Fig. 3 (c) is the most significant among these three graphs, for the gauge station of this graph is located in the front of the cliff where the flow structure is extremely complex. Wave rushes from the coastline in a shape of water jet. Then, it impacts and runs up on the cliff and falls back to the beach. Large quantity of air is entrained in water when backflow interacts with the incident flow. The velocities of water particles fluctuate violently due to the water-cliff interaction and the drastic water-air mixing. Intense spray of water makes it hard to measure the water elevation with a wave gauge. Hence, Sim et al. (2015) employed three HD Pro c910 web cameras to observe wave transformation



besides the Ultralab sensors. Data of video recordings are also presented in Fig. 3 (c) marked by × from Sim et al. (2015). It can be seen from Fig. 3 (c) that the crest value of video data is 18% smaller than the value of sensor data, which reveals that it is hard to determine the true trace of water surface in such a complex condition. The result of fine-grid is between the result of video data and sensor data, the result of middle-mesh is similar to the video data, and the result of coarse-mesh is 3%

5    smaller than video data. In general, our model shows a good performance in this verifiable example, even when the flow regime is extremely volatile. More verification of our model can been found in Zhao et al. (2014). However, for the coarse-grid has a little underestimation and the fine-grid has low time efficiency, the middle- grid will be adopted to complete the remaining case studies.

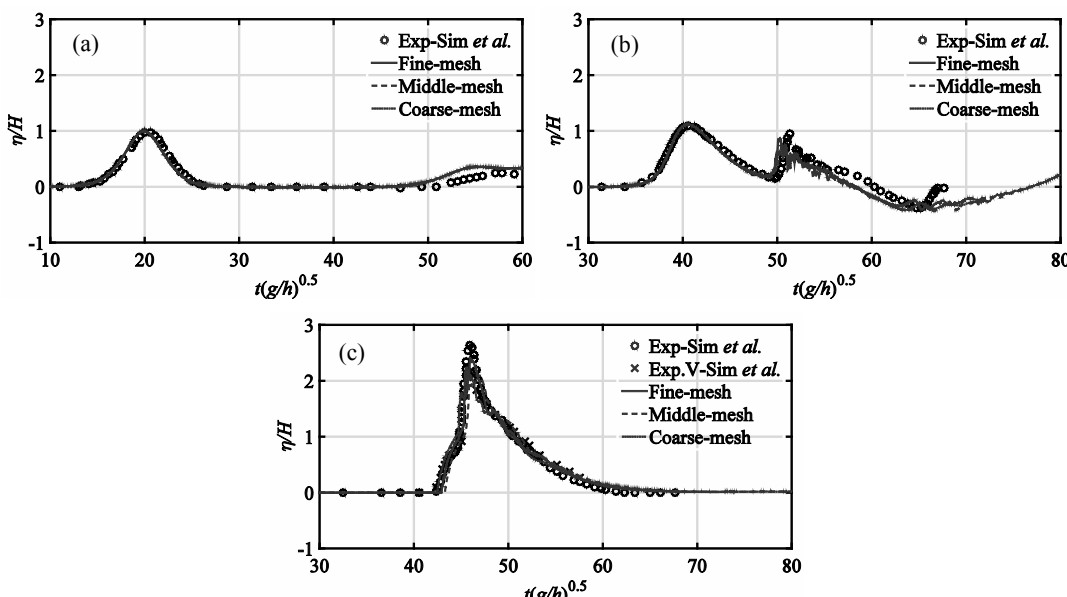

Fig. 3 Time series of experimental data and predicted water elevations using different grids: (a) S1, (b) S2, (c) S3.

### 3.3 The tsunami amplification factor in 1# tank

Fig. 4 describes the results of the tsunami wave amplification factor in 1# tank. The vertical coordinates are $H_m/H_r$, in which $H_m$ means the local wave height and $H_r$ means the reference wave height at a reference location $x_r$. The reference location in

15    present study is $x_r$= -0.87 m (same as Sim et al., 2015), and the reference wave height $H_r$ is provided by present numerical result. From the results in Fig. 4, it is observed that there exists a critical cliff slope about $\theta_4$=45° . When the angle of cliff slope is smaller than the critical angle, the tsunami wave amplification factor increases with the increase of the cliff slope angle. When the slope is steeper than the critical slope, the effect of the cliff slope becomes insignificant. This result is similar to Sim et al. (2015). As for Fig. 4 (e) and (f), the wave gauges are close to the cliff, when the cliff slope is gentle,

20    close to 22°, the tsunami amplification factor increases with the decrease of the incident wave height. It is noteworthy that under the condition of steep cliff, the tsunami amplification factor increases with the increase of the incident wave height, in





contrast to the gentle cliff. As the cliff slope becomes steep, the differences between different incident waves become small firstly. The critical cliff slope is about $\theta_4$=45$^o$. When the angle of cliff slope is greater than 45$^o$, the differences start to get bigger again. A possible reason for this overturned phenomenon is that the velocity of water particle in the high wave is faster, which allows the high wave easier to run up on the cliff. So that when the cliff slope is gentle, the water of high wave

5    rushes along the cliff and reaches a rearward area, instead of accumulating in the front of cliff as water of small wave cases does. Then, as the cliff slopes get steep, the so-called rearward area becomes hard to reach for the high wave. This change makes the high wave to accumulate water in the area of these two gauge stations. Moreover, the tsunami amplification factor at these two stations keeps increasing with the increase of the cliff slope angle for a given original incident wave no matter the angle is greater or smaller than 45$^o$, which is different to the results of other four stations mentioned before. Hence, the

10   presence of a cliff does amplify the water elevations on the beach. The influence is particularly evident for the high wave. In present study, the largest tsunami amplification factor is 2.86, as shown in Fig. 4 (f).

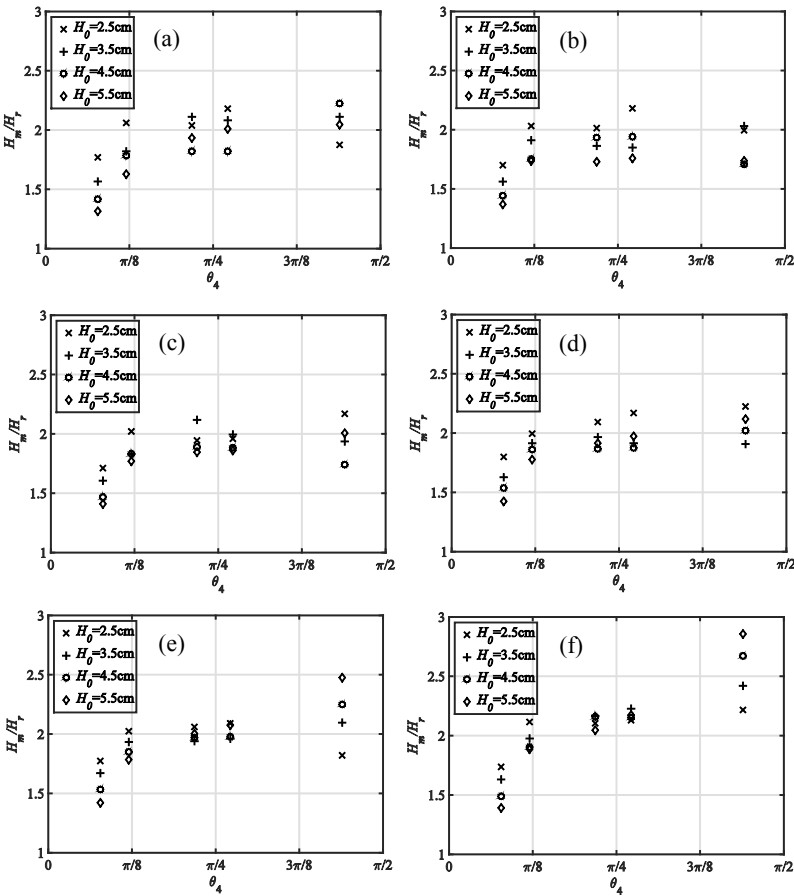

15    Fig. 4 Wave amplification factors, $H_m/H_r$ of different cliff angles: (a) $x$ = 0.0 m, (b) $x$ = 0.06 m, (c) $x$ = 0.11 m, (d) $x$ = 0.13 m, (e) $x$ = 0.16 m, (f) $x$ = 0.21 m



### 3.4 Time evolution of relative wave elevation in 2# tank

Fig. 5 depicts the time series of relative wave elevation in 2# tank for the cases 1-12. Here, four submarine gentle slope lengths and three incident wave heights are considered. The predicted results at $x = 0$ m are shown in Figs. 5 (a), (c), (e), whereas at $x = 0.4$ m shown in Figs. 5 (b), (d), (f). It can be noticed in Figs. 5 (a), (c), (e) that there are conspicuous

5    distinctions between the incident and the reflected wave. The relative wave height of incident wave increases with the decrease of the original wave height. Due to the shoaling, the length of the submarine gentle slope has a remarkable effect to the waveform and arrival time. Long gentle slope makes the wave asymmetry more significant. In addition, the longer the gentle slope is, the later the wave arrives. The reflected wave fluctuates remarkably because of the complex flow pattern, but the crest is higher than the incident wave. As for Fig. 5 (b), (d), (f), the wave gauges are close to the cliff, it is hard to

10   distinguish the incident and reflected wave. The superposition of incident and reflected wave makes the crest of water level much higher than the results of Fig. 5 (a), (c), (e). The effect of the length of gentle slope to the relative wave height at this position is puny. The waveform and arrival time of case13~24 are similar to case1~12, which is not shown here.

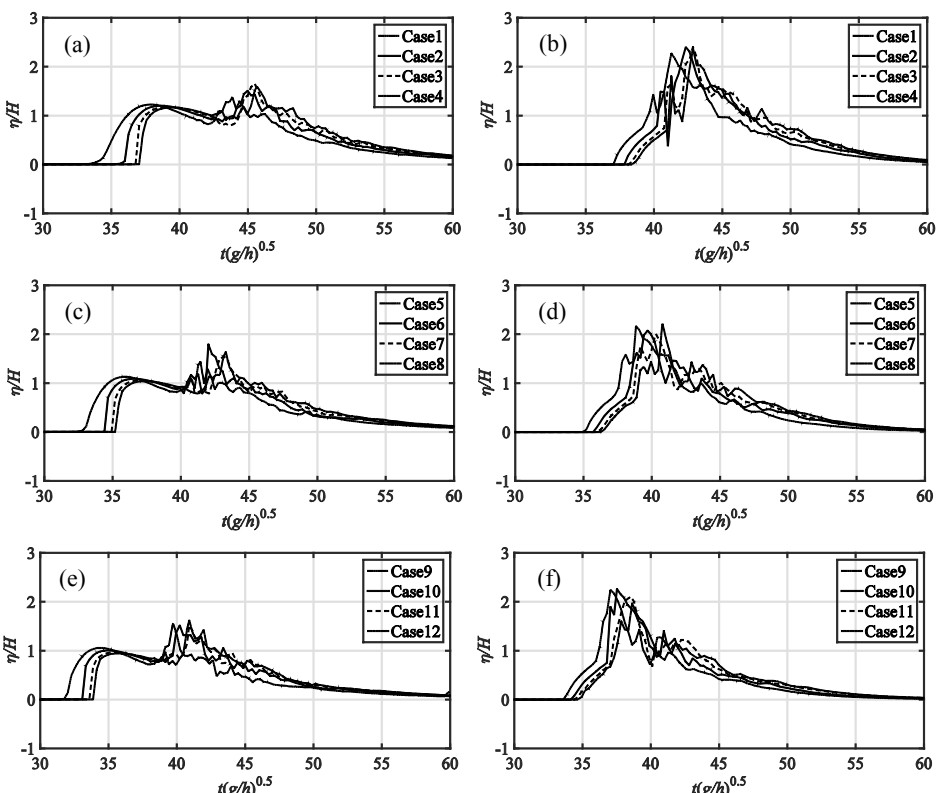

Fig. 5 Time series of relative wave elevation in 2# tank: (a), (c) and (e) $x = 0$ m; (b), (d) and (f) $x = 0.4$ m.




## 3.5 Relative wave height in front of the cliff in 2# tank

Maximum relative wave heights at five gauges: $x$=0m, 0.1m, 0.2m, 0.3m, 0.4m, in 2# tank are shown in Fig. 6. The predicted relative wave height is normalized to the incident wave height, $H$. The trend line is also presented as the black lines in Fig. 6. In Figs. 6 (a) and (b), the maximum relative wave heights are greater than 2.5, and the gradient of trend lines is large. In Figs

5   (c) and (d), the maximum relative wave height at each gauge is 2.4, the trend lines are not as steep as those in Figs. (a) and (b). In Figs. 6 (e) and (f), the trend lines are gentlest, especially when the cliff is toe-eroded. In summary, in the case of small wave height, the development of wave height is quicker, and finally a larger relative water height appears. This result reveals a moderate surface wave magnitude may cause enormous destruction in near-shore areas.

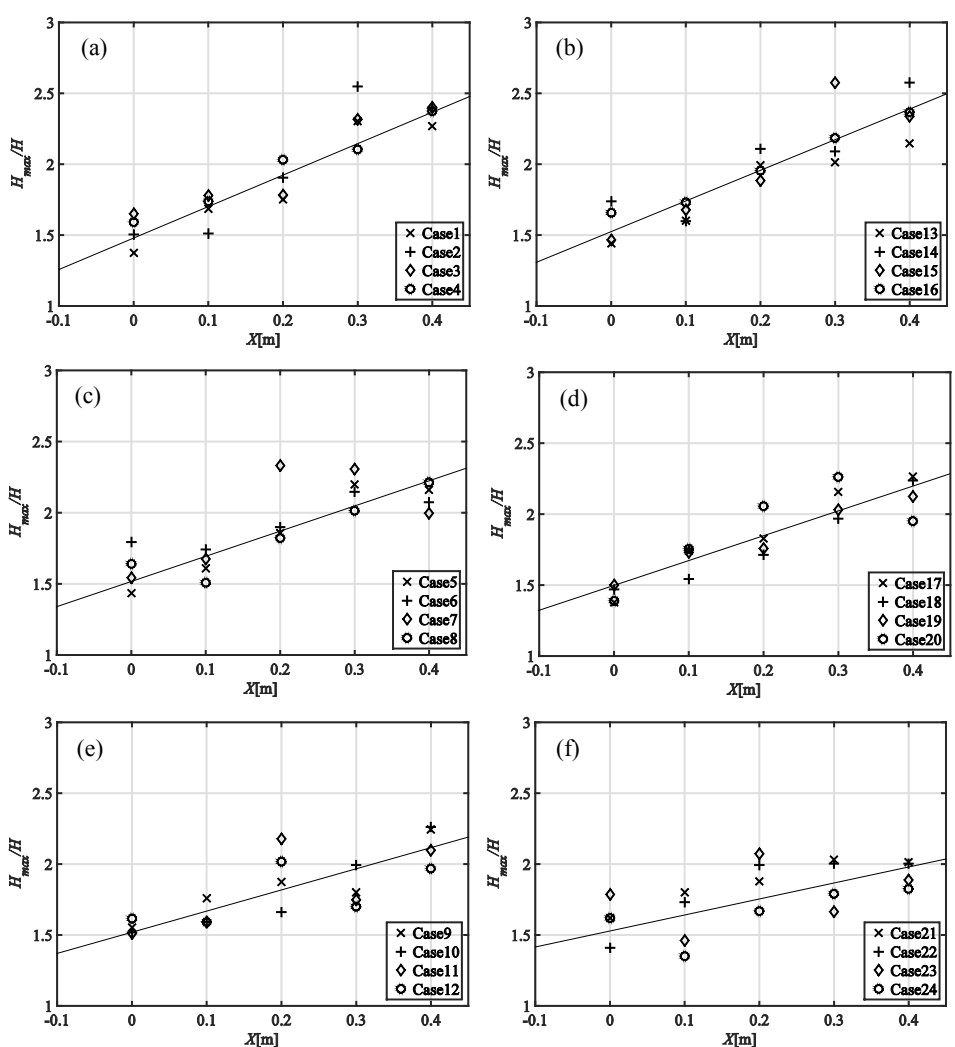

Fig. 6 Maximum relative wave height in front of the cliff in 2# tank: (a) and (b) $H = 0.04$ m, (c) and (d) $H = 0.05$ m, (e) and (f) $H = 0.06$ m.

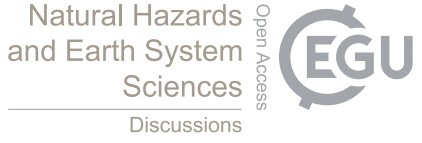

### 3.6 Wave run-up on the cliff in 2# tank

Fig. 7 displays the relative wave run-up on the cliff in 2# tank, in which different lengths of submarine gentle slope are used. The predicted run-up is normalized to the incident wave height, $H$. It can be seen in Fig. 7 that there exists a critical length of submarine gentle slope about $L$=2.292m. When $L < 2.292$ m, with a given cliff, the relative wave run-up increases

as the length of gentle slope increases. When $L$ goes over the critical length $L$=2.292m, the wave run-up for the case $H$ = 0.04 m decreases with the increase of the gentle slope length, but results of the case $H$ = 0.06 m still increase with the increase of the gentle slope length. Moreover, the main notable result is that the normal cliff gives an increase result and the toe-eroded cliff gives an opposite result for cases $H$ = 0.05 m. It reveals that the critical value relates to both the incident wave height and the inclination of a cliff. The increase of the incident wave height and the erosion at the cliff toe exaggerates

the critical value of submarine gentle slope length. As a whole, the run-up on a normal cliff is higher than that on a toe-eroded cliff. The maximum relative run-up on a normal cliff reaches up to 4.2. The possible reason is that the inclination of normal cliff is accordant with the direction of the incident wave while the toe-eroded cliff is contrary.

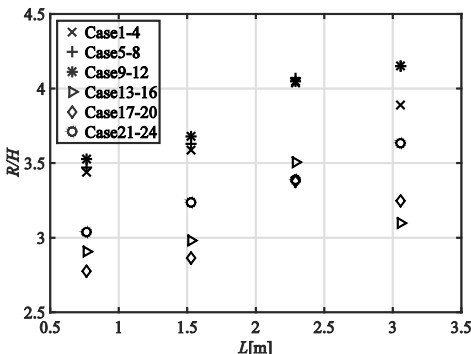

Fig. 7 Wave run-up on the cliff

### 3.7 Impact pressure analysis in 2# tank

Fig. 8 shows the time histories of impact pressure on the cliff at two pressure sensors P3 and P4. The detailed locations of pressure sensors are shown in Fig. 2(b). Figs. 8(a), (c), (e) , (g), (i) and (k) give the predicted results at location P3, whereas the results at location P4 shown in Figs. 8 (b), (d), (f), (h), (j) and (l). The recorded pressure is normalized to the hydrostatic pressure due to incident wave amplitude, $\rho gH$. The peak pressure can be divided into two categories: the early

peak and the later peak, which are caused by the wave impact and backflow, respectively. It is obvious that the high incident wave height is favorable for the appearance of early peak. Once the early peak appears, it is very larger comparing to the later peak. In the cases of small incident wave height, only later peak is observed at both measurement points. In the action period, the main scope of relative pressure ranges from 1.5 to 2.5, taking no account of the pressure peak.

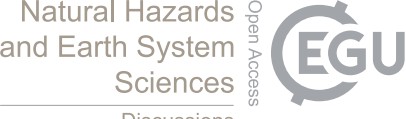









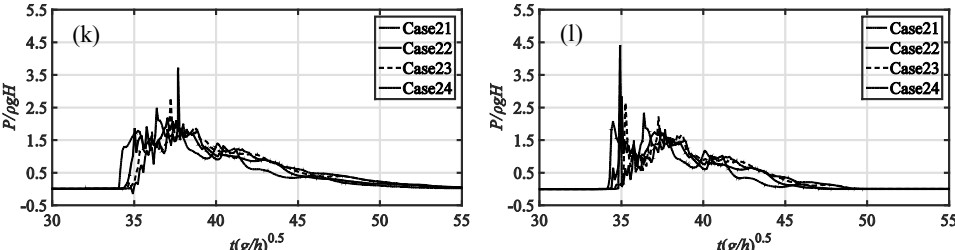

Fig. 8 Time histories of impact pressure on the cliff: (a), (c), (e) , (g), (i), (k) P3; (b), (d), (f), (h), (j), (l) P4.

The maximum pressures at five pressure sensors of all cases are also shown in Fig. 9. According to the analysis of Fig. 8, there is no peak when the maximum relative pressure is smaller than 2.5, here our attention will be paid on maximum relative pressure greater than 2.5. The inclination of a cliff affects the appearance of the pressure peak. Under the condition of a toe-eroded cliff, the generation of pressure peak is frequent. As for the normal cliff, the pressure peak is rare, but it can be terribly large once it appears. In addition, the pressure peak is found to be related to the length of submarine gentle slope $L$. When $L$ is small, it is easier for a small wave to generate an extreme pressure, which may be caused by backflow; and when $L$ becomes large, a high wave has a trend to generate an extreme pressure, which is probably caused by the direct wave impact.

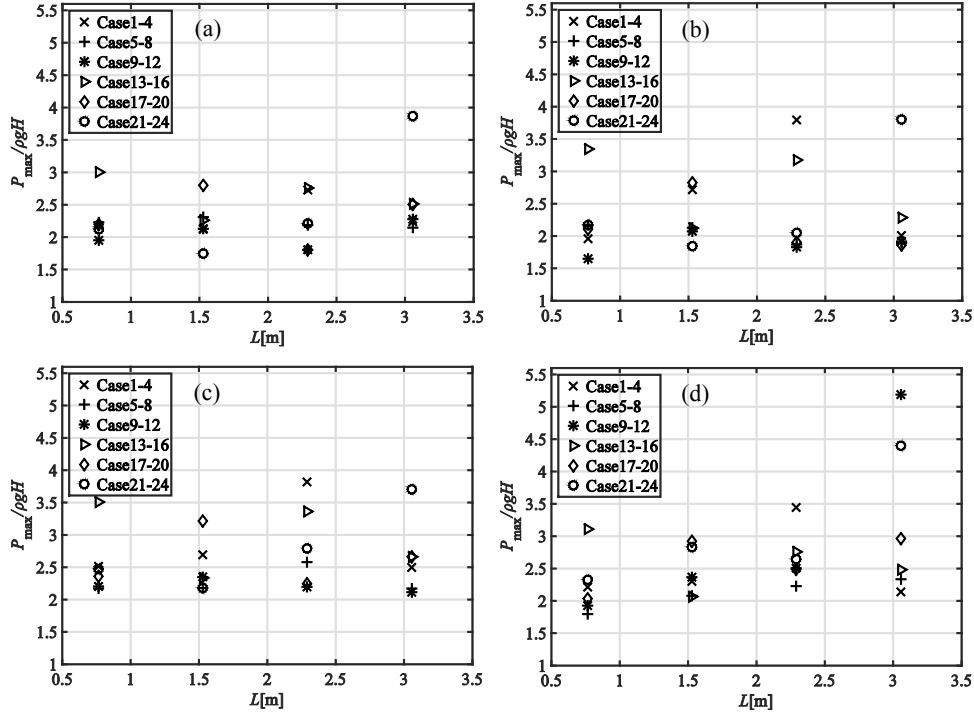





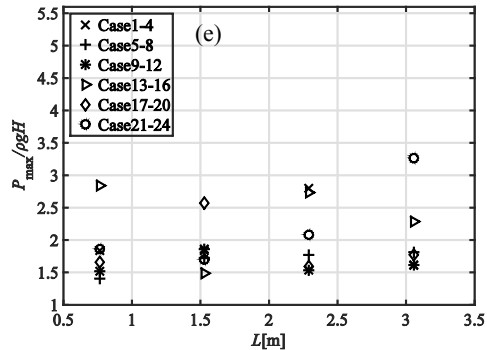

Fig. 9 Maximum impact pressure at five measuing points.

Fig. 10 demonstrates the snapshots of the wave impact, run up and fallback from the cliff and pressure distribution at different time instances. Two typical cases of a normal cliff $\theta_4 = 80.02°$ (case 12) and a toe-eroded cliff $\theta_4 = 91.91°$ (case 24) are simulated. The predicted results for the case 12 are shown in Figs.10 (a), (c) and (e), whereas case 24 shown in Figs. 10 (b), (d) and (f). Firstly, the wave approaches and impacts on the cliff at very onset, extreme pressure distribution appears in a small area near the toe of the cliff near the location of pressure sensor P4 (Figs. 10 (a) and (b)). Because of the cliff, the wave front is deviated and is deflected to a water jet along the cliff. As the water runs up the cliff and slowed down by the gravity action. Also it seems that the pressure impacting on the cliff has been decreased a little (Figs. 10 (c) and (d)). Moreover, significant water splashing of small water droplets can be observed, contributed from present accurate numerical model. Finally, the water falls back from the cliff due to gravity effect. This causes the formation of a backward plunging backflow hitting the underlying water, entrapping air. At this stage, extreme pressure appears again and the scope is large, mainly on the beach in front of the cliff. Due to the cliff inclination angle effect, the detailed water flow characteristic on the toe-eroded cliff is found to be different from that on the normal cliff. When the wave propagates along the gentle slope and hits the cliff, the water front horizontal velocity is changed upward along the cliff, as shown in Figs. 10 (a) and (b). As time passes by, the wave run-up on the toe-eroded cliff deviates from the cliff due to gravity effect (Fig. 10 (d)), which will cause a violent plunging breaker onto the underlying water (Fig. 10 (f)). While for a normal cliff, the wave runs up along the cliff, complex backflow is generated and later wave breaking and air-entrainment occurs between the backflow and the underlying water. A comparison of left column and right column of Fig. 10 reveals that although the general key flow features such as water breaking, water splashing and air-entrainment are not significantly affected by the inclination angle of the cliff, the detailed flow features is different. The above findings are partly consistent with the results of Huang et al. (2013). Combining with the results in Fig. 8, the extreme pressure caused by the direct impact is very large and extraordinarily concentrated, which may destroy the structures and human beings near the coast. While the backflow also produces extreme pressure with a widely affecting area. The worst thing is that it may carry plenty of floats from damaged construction and vegetation, causing secondary damage.





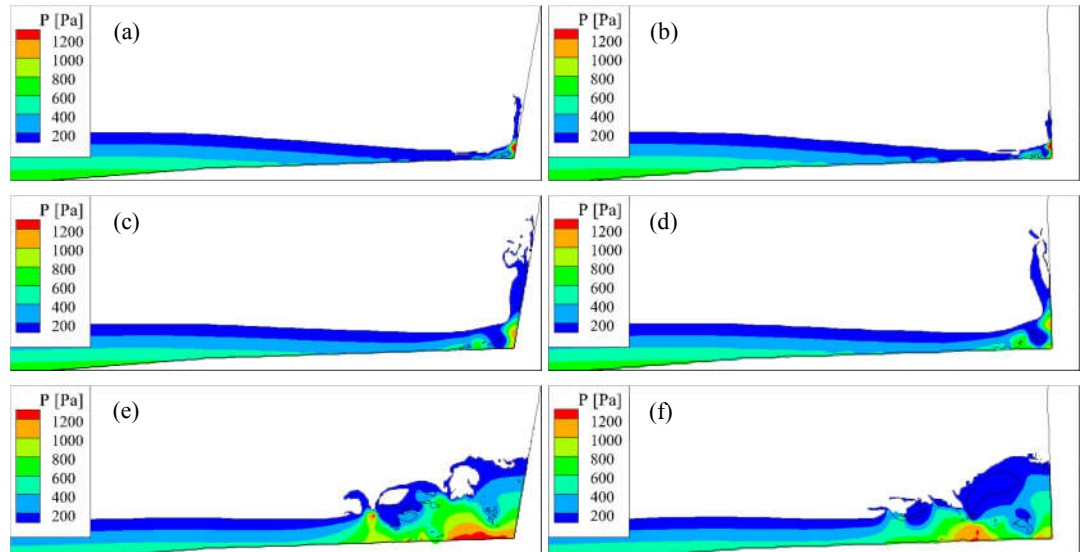

Fig. 10 Snapshots of impact pressure distribution in front of the cliff. (a), (c), (e) case12, (b), (d), (f) case24.

## 4 Conclusions

In this study, tsunami wave impact and run-up in the presence of submarine gentle slopes and a coast cliff are investigated numerically using a CIP-based model. Numerical results are firstly compared with available experimental data and the good agreement revealed the ability of our model to solve the complex flow field, such as wave breaking, water-air mixing and violent impact. The results can be summarized as follows.

(1) The angle of cliff slope has a critical value about 45°, different characteristics of tsunami amplification factor with different incident wave heights has been found when the angle is greater or smaller than 45°.

(2) The length of submarine gentle slope influences the tsunami wave arrival time and the run-up, and has a critical value about $L = 2.292$ m in this study.

(3) When wave transforms in front of cliff, the cases with small incident wave height has a large amplification factor, which means a devastating tsunami may be caused by a moderate source.

(4) It is easier for tsunami waves to run up on a normal cliff than on a toe-eroded one.

(5) There are two opportunities for the appearance of pressure peak during the process of tsunami wave run-up and impact. One is the direct impacting pressure when tsunami waves first hit the coastal cliff, and the other is caused by the backflow from the cliff after run-up with a widely affecting area.

The present study gives time history of tsunami evolution from open sea to coastal area, which is rare in field study. Several topographies and different incident waves has been considered. Comparing with the SWE result, which may underrate and need amendment, present results can simulate the tsunami in near shore areas more accurately. The present



model is helpful for tsunami forecast, dangerous prediction and post-disaster analysis. Furthermore, combining with geology knowledge, the earthquake source magnitude and generation location can be determined.

## Acknowlegements

This work was financially supported by the National Natural Science Foundation of China (Grant Nos. 51479175,

51679212), Zhejiang Provincial Natural Science Foundation of China (Grant No. LR16E090002).

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
