# Peer review of "A numerical study of tsunami wave impact and run-up on coastal cliffs using a CIP-based model"

_Natural Hazards and Earth System Sciences, 2017_

## Referee Comment (RC1) · Anonymous Referee #1 · 2 Mar 2017

The paper present a numerical study of the run up on cliffs located at the back of a beach. In order to do the experiments the authors propose the use of an "in-house" model based on VOF techniques and a numerical channel of 1 m long. In order to test the validity of the model, authors reproduce experiments published by Sim et al., 2015. In these experiments, solitary waves of different heights are propagated over four segments profiles (sea floor, continental slope, continental shelf, beach, and cliff) obtaining results accordingly to the previous observations. Next, Authors investigate the effect of cliff slope on the run up using the same channel configuration and varying the cliff slope. On a second channel setup, Authors investigate the effect of continental shelf gentle slope with four different solitary waves and two cliff slopes. Finally, some results are explained.

Although model results are worthy and validation with laboratory experiments are also

good in my opinion the paper is not suitable for publication as it is for the next reasons:

a) The paper applies an 'in-house' CIP model, probably a VOF. Although this is not completely clear, model description is very poor. For example is not clear what are the differences and advantages of using this model instead of a traditional VOF.

b) Regarding the experiments setup it is not stated the scale of the channel, moreover being numerical experiments, why experiments should be scaled? Considering the water depth of the channel (0.35 m) and a complete ocean profile (from the coast to deep ocean) a scale of the order of 1:10000 can be guessed. This lead to continental shelf length of 15 km and depth of 1 km (typical values for these are 80 km and 200 m). The use of these values must be justified. The wave heights used on simulations are one order of magnitude less than the ocean depth i.e., total depth h=0.35 and tsunamis waves H from 0.025 to 0.06 (For example a wave of 0.025 scaled to 1:10000 gives a wave of 250 m). This is completely unrealistic for tsunamis. In the case of laboratory experiments, some concessions must be done, but on numerical experiments there is no reason to do this. If there is a numerical or mathematical reason to scale the waves like this, it must be stated.

c) On section 3.4 where time evolution of wave is analyzed on tank #2, Authors find, as expected, that the longer the gentle slope is, the later the wave arrives. Since we are taking about long waves this is an obvious results and there is nothing new on the description.

d) On section 3.5, authors evaluate the relative wave height in front of the cliff on tank #2 at different distances. Data describes a linear trend, nevertheless authors do not mention the trend values, and do not intent to find an explanation. Furthermore, was possible to find a relationship between this trend and profile slope. None of these is done by authors.

e) On section 3.6, authors evaluate the run up on tank #2. They found that for larger continental platforms the run up is larger. This implies that (section 3.4) longer continental slopes, produce waves traveling slower and with higher run up. Authors do not discuss this idea or any other.

f) In conclusion, description of results is very poor, there is no discussions and conclusions are just a summary of the remarkable results.

g) Finally it is rather hard going and the English needs to be improved considerably.

---

## Author Response (AR1)

Dear editor:

Thank you very much for your kind consideration. We have revised the manuscript according to two reviewers' comments. The important modifications are listed as follows and are shown in red color in the revised manuscript.

**Lists of changes**

1. **Page 1, Title**

   The words "*run-up and impact*" were changed to "*impact and run-up*".

2. **Page 1,** the author's list was updated in the revision and the corresponding author was transferred to Dr Yangyang Gao.

3. **Page1, Abstract, line 8**

   The words "*lack of the understanding*" were changed to "*lack of understanding*".

4. **Page1, Abstract, line10-11**

   The expression

   "*named a Constrained Interpolation Profile (CIP)-based model in Zhejiang University (CIP-ZJU).*"

   was changed to

   "*a Constrained Interpolation Profile (CIP)-based model*".

5. **Page 1, Abstract, line 11-13**

   The three ',' in sentence

   "*The model employs a high-order finite difference method, the CIP method as the flow solver, utilizes a VOF-type method, the Tangent of hyperbola for interface capturing/Slope weighting (THINC/SW) scheme to capture the free surface, and treats the solid boundary by an immersed boundary method.*"

   were changed to ';' as follow

   "*The model employs a high-order finite difference method, the CIP method as the flow solver; utilizes a VOF-type method, the Tangent of hyperbola for interface capturing/Slope weighting (THINC/SW) scheme, to capture the free surface; and treats the solid boundary by an immersed boundary method.*"

6. **Page 1, Abstract, line 16**

   The words "*rule of*" were deleted.

7. **Page 1, Abstract, line 16**

   The words "*angle of cliff slope*" were changed to "*gradient of cliff slope*".

8. **Page 1, Abstract, line 17**

   The words "*critical angle*" were changed to "*critical value*".

9. **Page 1, Abstract, line 17**

   The words "*a toe-erosion cliff*" and "*a normal cliff*" were changed to "*toe-erosion cliff*" and "*normal cliff*".

10. **Page 1, Abstract, line 18**

    The words "*for most cases*" were inserted.

11. **Page 1, Introduction, line 22-23**

    The sentence

    "*Tsunami one of the coastal hazards in the world, can be caused by earthquake, volcanic eruption and submarine landslide.*"

was changed to

"*Tsunami is one of the most disastrous coastal hazards in the world, which can be caused by earthquake, volcanic eruption and submarine landslide.*"

12. **Page 2, Introduction, line 13**

    The words "*Sim et al., 2005*" were changed to "*Sim and Huang, 2015*".

13. **Page 2, Introduction, line 14**

    The word "*incompatible*" was changed to "*a problem*".

14. **Page 2, Introduction, line 17**

    The words "*time history*" were changed to "*time-series*".

15. **Page 2, Introduction, line 21-31**

    The sentences

    "*Finite difference method is widely used in various CFD models as flow solver. Hitherto, the accuracy of finite difference method is still a great challenge. In this paper, we introduce a CFD model based on constrained interpolation profile (CIP) algorithm. The CIP method was first introduced by Takewaki et al. (1985) as a high order method to solve the hyperbolic partial differential equation. Tanaka et al. (2000) proposed a new version of the CIP-CSL4, which overcomes the difficulty of conservative property. Hu et al. (2009) simulated strongly nonlinear wave-body interactions used a CIP-based Cartesian grid method, and the results were in good agreement with experiment data. Kawasaki et al. (2015) developed a tsunami run-up and inundation model based on CIP method, and high accurate water surface profile was observed by using slip condition on the wet-dry boundary. Fu et al. (2017) simulated the flow past an in-line forced oscillating square cylinder by a CIP-based model. CIP method can be not only applied in CFD, but also has good performance in other areas. Sonobe et al. (2016) employed CIP method to simulate sound propagation involving the Doppler-effect.*"
    were inserted.

16. **Page 2, Introduction, line 32**

    The word "*problem*" was inserted.

17. **Page 2, Introduction, line 32-34**

    The sentence

    "*Hirt et al. (1981) put forward a mass conservation method named Volume of Fluid (VOF).*"
    was expanded to

    "*Hirt et al. (1981) put forward a mass conservation method named Volume of Fluid (VOF), which is flexible and efficient for treating complicated free boundary configurations.*"

18. **Page 3, Introduction, line 1-8**

    The sentences

    "*Yokoi et al. (2013) proposed a numerical framework consisting of CLSVOF method, multi-moment methods and density-scaled CSF model. The framework can well capture free surface flows with complex interface geometries. More recently, conventional VOF has been widely used by combining with various additional scheme. Malgarinos et al. (2015) proposed an interface sharpening scheme on the basic of standard VOF method, which effectively restrained interface numerical diffusion. Gupta et al. (2016) used a coupled VOF and pseudo transient method to solve free surface flow problems, and the numerical solution was compared well with analytical or experimental data. Quiyoom et al. (2017) simulated the process of gas-induced liquid mixing in a shallow vessel, found that the mixing time predicted*"

*by EL+VOF was in good agreement with the measurements.*"

were inserted.

19. **Page 3, Introduction, line 9**

    The words "*of solid boundary*" were inserted.

20. **Page 3, Introduction, line 12**

    The sentence

    "*CFD is more convenient comparing with experimental data or field observation.*"

    was changed to

    "*CFD is more convenient and economical than laboratorial experiment and field observation.*"

21. **Page 3, Introduction, line 12-14**

    The sentence

    "*The most valuable achievement is that it can give time history data of waveform, pressure and flow velocity field to understand the evolution and impact mechanism of a tsunami in near-shore areas.*"

    was changed to

    "*The most attractive feature is that it can provide time-series data of waveform and flow field, which are helpful for a better understanding of the inundation mechanism of wave in near-shore areas.*"

22. **Page 3, Introduction, line 14-20**

    The sentences

    "*Markus et al. (2014) introduced a Virtual Free Surface (VFS) model, which enabled the simulations of fully submerged structures subjected to pure waves and combined wave–current scenarios. Vicinanza et al. (2015) proposed new equations to predict the magnitude of forces exerted by the wave on its front face. The equations were added in 5 random wave CFD model and good agreement was obtained when compared with empirical predictions.Oliveira et al. (2017) utilized PFEM to simulate complex solid-fluid interaction and free surface, so that a piston numerical wave-maker are implemented in a numerical wave flumes. Regular long wave was successfully generated in the numerical wave flumes.*"

    were inserted.

23. **Page 3, Introduction, line 21-25**

    The sentence

    "*When efficient numerical algorithms adopted, a CFD-based numerical simulation with a suitable scale can be applied to study the tsunami inundation in coastal areas.*"

    was expanded to

    "*When efficient numerical algorithm is adopted, CFD numerical simulation can be applied to study the tsunami inundation in coastal areas. However, limited by the computational efficiency and numerical dissipation of finite difference method, the quantity and slenderness ratio of computational grids should be moderate. Because of the high proportion of spatial span in horizontal and vertical in most cases, reasonable abnormal model scale in horizontal and vertical is necessary, similar to laboratorial experiment.*"

24. **Page 3, Introduction, line 26-Page 3, line4**

    The sentences

    "*The purpose of present work is to understand the characteristics of a tsunami, helping*

*inverse the generation mechanisms and provide reference for the tsunami forecasting and post-disaster treatment.*"

and the paragraph

"*Coastal cliffs are one of the common coastal landforms, representing approximately 75% of the world's coastline (Rosseret al., 2005), such as the coast of Banda Aceh in Indonesia and the steep slope at San Martin (António et al., 1993). The existence of cliff can not only influence the impact and run-up of a tsunami wave but also the erosion-deposition. Different layers provide variations in resistance to erosion (Stephenson et al., 2011). Particularly, some coastal cliffs consisting of soft rocks are eroded at the toe (Yasuhara et al., 2002), which makes it easier to be destroyed. It is indispensable to understand the function of coastal cliffs and submerged gentle slopes during the tsunami wave approaches near shore. In this study, tsunami wave run-up and impact on coastal cliffs is simulated using an in-house code, named the CIP-ZJU model. Considerable attention is paid to the influence of different coastal topographies on the tsunami inundation in near shore areas. Steep cliffs on the beach and submerged gentle slopes are considered. The submerged gentle slope such as the continental shelf, affects the waveform evolution and wave celerity before tsunami waves reach the shoreline. Tsunami amplification factor, relative wave height, run-up on the cliff and impact pressure will be analyzed in this work.*"

were rearranged as

"*The purpose of present work is to understand the characteristics of a tsunami, helping inverse the generation mechanisms and provide reference for the tsunami forecasting and post-disaster treatment. In this study, tsunami wave impact and run-up on coastal cliffs is simulated using an in-house code, named the CIP-Based model. Considerable attention is paid to the influence of different coastal topographies, steep cliffs on the beach and submerged gentle slopes are considered. Coastal cliffs are one of the common coastal landforms, representing approximately 75% of the world's coastline (Rosser et al., 2005), such as the coast of Banda Aceh in Indonesia and the steep slope at San Martin (António et al., 1993). The existence of cliff can not only influence the impact and run-up of a tsunami wave but also the erosion-deposition. Different layers provide variations in resistance to erosion (Stephenson et al., 2011). Particularly, some coastal cliffs consisting of soft rocks are eroded at the toe (Yasuhara et al., 2002), which makes it easier to be destroyed. It is indispensable to understand the function of coastal cliffs and submerged gentle slopes during the tsunami wave approaches near shore. The submerged gentle slope such as the continental shelf, affects the waveform evolution and wave celerity before tsunami waves reach the shoreline. Tsunami amplification factor (Satake, 1994), relative wave height, run-up on the cliff and impact pressure will be analyzed in this work.*"

25. **Page 4, Introduction, line 5-6**

    The sentence

    "*Section 3 presents the initial condition and model validation;*"

    was changed to

    "*Section 3 provides the initial condition and numerical wave-maker; Section 4 presents model validation;*"

26. **Page 4, Introduction, line 7-8**

    The sentence

"*finally, the conclusion is made in Section 4.*"

was changed to

"*finally, the discussion and conclusion are made in Section 5 and 6, respectively.*"

27. **Page 4, Section 2, line 23**

    The subsection "*2.2 Numerical methods*" was changed to "*2.2 The fractional step approach*".

28. **Whole text**

    The expression "*equation (\*)*" was changed to "*Eq. (\*)*".

29. **Page 5, Section 2.2, line 4**

    The sentences

    "*The CIP method was first introduced by Takewaki et al. (1985) as an efficient method to solve the hyperbolic partial differential equation. The basic principle of CIP is an interpolation in a grid by a cubic polynomial with the value and differential on the grid node. The advantage of CIP is that it can provide a third order interpolation function in a single grid, which makes it a compact high order scheme.*"

    were deleted.

30. **Page 5, Section 2.2, line 14**

    The sentence

    "*The THINC/SW was put forward by Xiao et al. (2011), using a hyperbolic tangent function with adjustable parameter to interpolate.*"

    was deleted.

31. **Page 5, Section 2, line 16-Page6, line 18**

    The following subsection was inserted, the corresponding following subsections' number, equations' number and figures' number were changed.

    "*2.3 CIP method*

    *The basic principle of CIP is that when computing the advection of a variable f, both the transportation equation of f and the transportation equation of its spatial gradient* $g = \partial f / \partial x$ *are used (seen in Fig. 1).*

    *To explain the step of CIP method, we take the following 1D advection equation as an example.*

    $$\frac{\partial f}{\partial t} + u\frac{\partial f}{\partial x} = 0 \tag{9}$$

    *By differentiating Eq. (9) with respective to x, we have the equation about the spatial derivative as*

    $$\frac{\partial g}{\partial t} + u\frac{\partial g}{\partial x} = -g\frac{\partial u}{\partial x} \tag{10}$$

    *where* $g = \partial f / \partial x$. *For simplicity, we assume a constant advection velocity. Then Eq. (10) has the same form as Eq. (9). For the case of u > 0, we may approximate a profile for $f^n$ inside the upwind cell $[x_{i-1}, x_i]$ as*

    $$F_i^n(x) = a_i(x - x_i)^3 + b_i(x - x_i)^2 + c_i(x - x_i) + d_i \tag{11}$$

    *Since, the spatial gradient of $f^n$ can be written as*

    $$G_i^n(x) = a_i(x - x_i)^2 + b_i(x - x_i) + c_i \tag{12}$$

    *As shown in Fig. 2 the profile at the time step n+1 is obtained by shifting the profile with –u* $\triangle t$*, i.e., the time evolution of the function f and g can be obtained by using the following Lagrangian invariants.*

$$f_i^{n+1} = F_i^n(x_i - u \cdot \Delta t) \quad (13)$$

$$g_i^{n+1} = G_i^n(x_i - u \cdot \Delta t) \quad (14)$$

*Therefore we call the CIP scheme as a Semi-Lagrangian method.*

[Figure]

*Fig. 2 CIP scheme as a kind of Semi-Lagrangian method*

*The four unknown coefficients in Eq. (11) can be determined by using known quantities $f_i^n$, $f_{i-1}^n$, $g_i^n$, $g_{i-1}^n$. It is not difficult to obtain*

$$a_i = \frac{g_i^n - g_i^n}{\Delta x^2} - \frac{2(f_i^n - f_{i-1}^n)}{\Delta x^3}, \quad c_i = g_i^n$$

$$b_i = \frac{2g_i^n + g_i^n}{\Delta x} - \frac{3(f_i^n - f_{i-1}^n)}{\Delta x^2}, \quad d_i = f_i^n \quad (15)$$

*By introducing the spatial gradient of variable f, CIP can provide a third order interpolation function in a single grid. Comparing to traditional upwind schemes, CIP method has not only sub-cell resolution, but also compact high order structure.*"

**32. Page 6, Section 2, line 19-Page7,line 10**

The following subsection was inserted, the corresponding following subsections' number, equations' number and figures' number were changed.

"*2.4 THINC/SW scheme*

*The THINC/SW scheme, first put forward by Xiao et al. (2011), is used for free surface capturing of incompressible flows. Some test examples has indicated that the scheme has the features we need: mass conservation, a lack of oscillation (Ji et al. 2013). The basic idea of THINC/SW is for the profile of a volume function $\phi$ ($0 \le \phi \le 1$), a hyperbolic tangent function with adjustable parameter is used to interpolate inside an upwind cell, which is shown as follow:*

$$\chi_{x,i} = \frac{\alpha}{2}\{1 + \gamma \tanh[\beta(\frac{x - x_{i-1/2}}{\Delta x_i} - \delta)]\} \quad (16)$$

*where $\alpha$, $\gamma$, $\delta$, $\beta$, are parameters to be specified. $\alpha$ and $\gamma$ are used to avoid interface smearing, which are given by:*

$$\alpha = \begin{cases} \overline{\phi}_{i+1} & \text{if } \overline{\phi}_{i+1} \ge \overline{\phi}_{i-1} \\ \overline{\phi}_{i-1} & \text{otherwise} \end{cases} \quad (17)$$

$$\gamma = \begin{cases} 1 & \text{if } \overline{\phi}_{i+1} \ge \overline{\phi}_{i-1} \\ -1 & \text{otherwise} \end{cases} \quad (18)$$

*Parameter $\delta$ is used to determine the middle point of the hyperbolic tangent function, and is solved by:*

$$\frac{1}{\Delta x_i}\int_{x_{i-1/2}}^{x_{i+1/2}} \chi_i(x)dx = \overline{\phi}_i^n \quad (19)$$

*Parameter $\beta$ determines the steepness of the jump in the interpolation function varying from 0 to 1. In traditional THINC, a constant $\beta = 3.5$ is usually used which may result in ruffling the interface which aligns nearly in the direction of the velocity. To avoid this problem, in THINC/SW, $\beta$ is determined adaptively according to the orientation of the interface. In a 2D case, $\beta$ can be written as:*

$$\begin{cases} \beta_x = 2.3 \, | \, n_x \, | +0.01 \\ \beta_y = 2.3 \, | \, n_y \, | +0.01 \end{cases} \tag{20}"$$

33. **Page 7, Section 3, line 11**

    The Section "*3 Simulation setup*" was inserted, corresponding following Sections' number was changed.

34. **Page 7, Section 3, line 12**

    The Subsection "*3.1 Numerical wave tank*" was changed to "*3.1 Initial condition*".

35. **Whole text**

    The expression "*1# tank*" and "*2# tank*" were changed to "*Tank 1*" and "*Tank 2*", respectively.

36. **Page 7, Section 3, line 19-20**

    The sentence

    "*1# tank is used for verifying the accuracy of our model and investigating the effect of the angle of cliff slope and incident wave height on the tsunami amplification factor.*"

    was changed to

    "*Tank 1 is used for verifying the accuracy of our model and investigating cliff slope gradient and incident wave height, which may influence the tsunami amplification factor.*"

37. **Page 7, Section 3, line 20**

    The sentence

    "*The angles of slopes are $\tan\theta_1$= 25/17, $\tan\theta_2$= 1/15, $\tan\theta_3$= 1/30.*"

    was deleted

38. **Page 7, Section 3, line 21**

    The words "*in this tank*" were deleted

39. **Page 7, Section 3, line 21-22**

    The sentence

    "*Solitary waves are used as initial condition in the numerical modelling of tsunami.*"

    was changed to

    "*Solitary wave is used as an analogue of tsunami in the numerical modelling.*"

40. **Page 7, Section 3, line 22-24**

    The sentence

    "*S1~3 in Fig. 1 are three measuring gauges of water elevation for the comparison of numerical result and experimental data, located at x= -7.67 m, -0.87 m and 0.11 m, respectively.*"

    was changed to

    "*S1~3 in Fig. 3 are three gauges of water elevation, located at x= -7.67 m, -0.87 m and 0.11 m, respectively. Outcomes of S1~3 are used for the comparison between numerical results and experimental data (Sim, 2017).*"

41. **Page 7, Section 3, line 24**

    The sentence

    "*Besides, six measuring gauges of water level elevation in front of the cliff are employed to*

*analyze the tsunami amplification factor,"*

was changed to

*"Besides, six gauges of water elevation are employed to calculate the tsunami amplification factor,"*

42. **Page 7, Section 3, line 25**

   The words "*not draw in Fig. 1*" were changed to "*not drawn in Fig. 3*".

43. **Page 8, Section 3, line 6**

   The sentence

    "*The angles of slopes are $\tan\theta_1$ = 1.38, $\tan\theta_2$ = 0.08, $\tan\theta_3$ = 0.02.*"

   was deleted

44. **Page 8, Section 3, line 7**

   The word "*set*" was changed to "*used*".

45. **Page 8, Section 3, line 8-9**

   The sentence

   "*Two kinds of cliff: normal cliff $\theta_4$ = 80.02$^o$, and toe-eroded cliff $\theta_4$ = 91.91$^o$ are considered.*"

   was changed to

   "*Two kinds of cliff, normal cliff of $\theta_4$ = 80.02$^o$ degree and toe-eroded cliff of $\theta_4$ = 91.91$^o$ degree are considered.*"

46. **Page 8, Section 3, line 9**

   The words

    "*Six measuring points of water level elevation*"

   were changed to

   "*Six gauges of water elevation*"

47. **Page 8, Section 3, line 11-12**

   The sentence

    "*The scales of these two kinds of wave tanks are same to the previous works of (Huang et al., 2013; Sim et al., 2015; Sim, 2017).*"

   Was inserted.

48. **Page 9, Section 3, line 1-Page 10, line 3**

   The following subsection was inserted, corresponding following subsections' number, equations' number and figures' number were changed.

   "*3.2 Numerical wave-maker*

   *By declaring a velocity of water particle in the left-most grid and assigning it a value from laboratory wave-paddle velocity, a numerical paddle wave maker is set at the left side of wave tank (Fig. 3 and Fig. 4 (a)).*

   *For a solitary wave, the approximate solution of wave profile near the wave paddle can be described as follow (Boussinesq, 1872):*

$$\eta = H \operatorname{sech}^2[\sqrt{\frac{3H}{4h^3}}(ct - \xi)] \qquad (21)$$

$$c = \sqrt{g(h + H)} \qquad (22)$$

   *where H, h, c, $\xi$ are the amplitude of the solitary wave, still water depth, wave celerity and wave-paddle trajectory, respectively.*

   *The wave-paddle velocity can be calculated as:*

$$u_1(\xi,t) = \frac{d\xi}{dt} \tag{23}$$

*For a long wave, the depth-averaged horizontal velocity of water particle derived from continuity equation is expressed as (Mei, 1983):*

$$u_2(x,t) = \frac{c\eta(x,t)}{h+\eta(x,t)} \tag{24}$$

*The horizontal water particle velocity adjacent to the paddle is equal to the wave-paddle velocity, which means that when $x = \xi$ in Eq. (24), $u_1 = u_2$.*

*Using Eqs. (24), (26) and (27), wave-paddle trajectory can be derived as an implicit expression:*

$$\xi(t) = \sqrt{\frac{4H}{3h}}h\tanh[\sqrt{\frac{4H}{3h^3}}(ct-\xi)] \tag{25}$$

*The stroke length of wave-paddle can be calculated as:*

$$S = \xi(\infty) - \xi(-\infty) = \sqrt{\frac{16H}{3h}}h \tag{26}$$

*In theory, period of solitary wave is infinite. In the application, it can be approximately define as follow:*

$$\tanh[\sqrt{\frac{4H}{3h^3}}(c\frac{T}{2}-\frac{S}{2})] = 0.999 \tag{27}$$

$$T = \frac{2}{c}\sqrt{\frac{4h^3}{3H}}(3.8+\frac{H}{h}) \tag{28}$$

*The water particle velocity imposed in the left-most grid can be given by:*

$$\bar{u}(t) = \frac{c\eta(\xi,t)}{h+\eta(\xi,t)} \quad 0 \le t \le T \tag{29}$$

*In our model, the left-most grid is not moveable as laboratory wave-paddle be, modification should be provided to Eq. (29). By some numerical tests, it is finally determined as:*

$$\bar{u}(t) = \frac{c[2\eta(\xi(t),t)-\eta(\xi(0),t)]}{h+\eta(\xi(0),t)} \quad 0 \le t \le T \tag{30}"$$

49. **Page 10, Section 4 line 7**

    The words "*Sim et al. 2015*" were changed to "*Sim, 2017*".

50. **Page 10, Section 4 line 7**

    The words "*cliff slope angle*" were changed to "*cliff slope gradient*".

51. **Page 10, Section 4 line 9**

    The words "*grid number*" were changed to "*grid quantity*".

52. **Page 10, Section 4 line 11**

    The words in Table 2

    "*Grid number in x direction*", "*Grid number in y direction*", "*Minimum grid size in x direction*", "*Minimum grid size in y direction*"

    were changed to

    "*Horizontal grid quantity*", "*Vertical grid quantity*", "*Horizontal minimum grid*", "*Vertica minimum grid*".

53. **Page 10, Section 4 line 13**

    The words "*Sim et al., 2015*" were changed to "*Sim, 2017*".

54. **Page 10, Section 4 line 20**

The words "*flow structure*" were changed to "*flow field*".

55. **Page 10, Section 4 line 21-23**

The sentences

"*Then, it impacts and runs up on the cliff and falls back to the beach. Large quantity of air is entrained in water when backflow interacts with the incident flow.*"

were expanded to

"*Then, the water jet impacts the cliff, accompanied by large pressure acting on the toe of cliff. Great acceleration is produced by the impact, making the water run up on the cliff. Under the action of gravity, water finally falls back, large quantity of air is entrained in water when backflow interacts with the incident flow.*"

56. **Page 10, Section 4 line 25**

The words "*Sim et al. (2015)*" were changed to "*Sim (2017)*".

57. **Page 11, Section 4 line 4-5**

The sentence

"*Data of video recordings are also presented in Fig. 3 (c) marked by × from Sim et al. (2015).*"

was changed to

"*Data of video recordings from Sim (2017) are also presented in Fig. 5 (c) marked by ×.*"

58. **Page 11, Section 4 line 9**

The words "*volatile*" were changed to "*unstable*".

59. **Page 11, Section 4 line 9-10**

The words "*However, for the*" were changed to "*However, as the*"

60. **Page 11, Section 4 line 13-14**

The sentence

"*Tsunami amplification factor is defines as a ratio of the local tsunami height to the tsunami height at a reference location.*"

was inserted.

61. **Page 11, Section 4 line 16**

The words "*same as Sim et al., 2015*" were changed to "*same as Sim, 2017*".

62. **Page 11, Section 4 line 17**

The words "*there exists*" were changed to "*there is*".

63. **Page 11, Section 4 line 17-18**

The sentence

"*When the angle of cliff slope is smaller than the critical angle*"

was changed to

"*When cliff slope is gentler than the critical value*"

64. **Page 11, Section 4 line 18**

The words "*cliff slope angle*" were changed to "*cliff slope gradient*"

65. **Page 11, Section 4 line 19**

The words "*cliff slope*" were changed to "*cliff slope gradient*"

66. **Page 11, Section 4 line 19**

The words "*Sim et al. (2015)*" were changed to "*Sim (2017)*"

67. **Page 11, Section 4 line 20-21**

The sentence

"*As for Fig. 4 (e) and (f), the wave gauges are close to the cliff,*"

was changed to

"*As for the influence of incident wave height, it can be found in Figs. 6 (e) and (f), of which the wave gauges are close to the cliff.*"

68. **Page 11, Section 4 line 21**

The words "$22^o$" was changed to "$\theta_4=22^o$"

69. **Page 12, Section 4 line 4**

The sentence

"*Fig. 4 Wave amplification factors, Hm/Hr of different cliff angles:*"

was changed to

"*Fig. 6 Wave amplification factors, Hm/Hr of different cliff:*"

70. **Page 12, Section 4 line 6-8**

The sentences

"*It is noteworthy that under the condition of steep cliff, the tsunami amplification factor increases with the increase of the incident wave height, in contrast to the gentle cliff. As the cliff slope becomes steep, the differences between different incident waves become small firstly. The critical cliff slope is about $\theta_4=45^o$. When the angle of cliff slope is greater than $45^o$, the differences start to get bigger again.*"

were changed to

"*As the cliff slope becomes steeper, the effect of different incident waves become negligible at first, and then become important. The critical cliff slope is about $\theta_4=45^o$. It is noteworthy that under the condition of steep cliff, the tsunami amplification factor increases with the increase of the incident wave height, in contrast to the gentle cliff.*"

71. **Page 12, Section 4 line 9**

The words "*overturned phenomenon*" were changed to "*contrary phenomenon*"

72. **Page 12, Section 4 line 9**

The word "*faster*" was changed to "*higher*"

73. **Page 12, Section 4 line 10-11**

The sentence

"*So that when the cliff slope is gentle, the water of high wave rushes along the cliff and reaches a rearward area, instead of accumulating in the front of cliff as water of small wave cases does.*"

was changed to

"*So that when the cliff slope is gentle, the water of high wave rushes along the cliff and reaches a rearward area, but water of small wave accumulates in the front of cliff.*"

74. **Page 13, Section 4 line 3**

The sentence

"*which is different to the results of other four stations mentioned before.*"

was deleted.

75. **Page 13, Section 4 line 5-6**

The sentence

"*It is similar with the result of Sim (2017), which has a value of 2.8.*"

was inserted.

76. **Page 13, Section 4 line 12**

The word "*Here*" was deleted.

77. **Page 12, Section 4 line 15-16**

The sentence

"*The relative wave height of incident wave increases with the decrease of the original wave height.*"

was changed to

"*The relative height of incident wave at x = 0 m increases with the decrease of the initial wave height.*"

78. **Page 13, Section 4 line 3**

The sentences

"*Due to the shoaling, the length of the submarine gentle slope has a remarkable effect to the waveform and arrival time. Long gentle slope makes the wave asymmetry more significant. In addition, the longer the gentle slope is, the later the wave arrives.*"

were deleted.

79. **Page 13, Section 4 line 16-17**

The sentence

"*The reflected wave fluctuates remarkably because of the complex flow pattern, but the crest is higher than the incident wave.*"

was changed to

"*The reflected wave fluctuates remarkably because of the complex flow pattern, and the crest reflected wave is higher than the incident wave.*"

80. **Page 13, Section 4 line 18**

The words "*of water level*" were deleted.

81. **Page 14, Section 4 line 5-6**

The sentence

"*The effect of the length of gentle slope to the relative wave height at this position is puny.*"

was changed to

"*The effect of initial wave height and length of submarine gentle slope is hard to find from Fig. 7, which remain to the following analysis.*"

82. **Page 14, Section 4 line 6**

The words "*x=0m, 0.1m, 0.2m, 0.3m, 0.4m*" were changed to "*x=0m, 0.1m, 0.2m, 0.3m and 0.4m*".

83. **Page 14, Section 4 line 9**

The words "*Maximum relative wave heights*" were changed to "*Wave heights*".

84. **Page 14, Section 4 line 9**

The words "*Maximum relative wave heights*" were changed to "*Wave heights*".

85. **Page 14, Section 4 line 10-Page 15, line 5**

The sentences

"*In Figs. 6 (a) and (b), the maximum relative wave heights are greater than 2.5, and the gradient of trend lines is large. In Figs (c) and (d), the maximum relative wave height at each gauge is 2.4, the trend lines are not as steep as those in Figs. (a) and (b). In Figs. 6 (e) and (f), the trend lines are gentlest, especially when the cliff is toe-eroded.*"

were changed to

"*In Figs. 8 (a) and (b), the maximum relative wave heights are greater than 2.5, and the gradients of trend lines are 2.22 and 2.16, respectively. In Figs. 8 (c) and (d), the maximum relative wave height is 2.4, the trend lines have gradients of 1.77 and 1.75, not as steep as those in Figs. 8 (a) and (b). In Figs. 8 (e) and (f), the trend lines are gentle, with gradients of 1.46 and 1.13.*"

86. **Page 15, Section 4 line 4**

    The words "*Figs (c) and (d)*" was changed to "*Figs. 8 (c) and (d)*"

87. **Page 15, Section 4 line 5-7**

    The sentence

    "*In summary, in the case of small wave height, the development of wave height is quicker, and finally a larger relative water height appears.*"

    was changed to

    "*In summary, in the case of smaller wave height, the development of wave height along with the decrease of distance to the cliff is more obvious, and finally a larger relative water height appears.*"

88. **Page 15, Section 4 line 5-7**

    The sentences

    "*As for lager wave, rate of wave height increase is very small, especially when the cliff is toe-eroded. The possible cause of this interesting phenomenon can be explained as follow. According to the result of Fig. 7, the crest of wave elevation is produced by the mixing of incident and reflected wave. Under the condition of large wave, the reflected wave is very strong, which makes the mixing occupy a wide area on the beach. As a result, energy distribution of large wave is not as concentrated as small wave be. The energy concentration helps the small wave to produce a higher relative wave height near the cliff.*"

    were inserted.

89. **Page 15, Section 4 line 13-Page16, line 5**

    The words "*Figs. 5(a), (c), (e)*" and "*Figs. 5(b), (d), (f)*" were changed to "*Figs. 7 (a), (c) and (e)*" and "*Figs. 7 (b), (d) and (f)*", respectively.

90. **Page 15, Section 4 line 15**

    The word "*relative*" was deleted.

91. **Page 15, Section 4 line 18**

    The sentences

    "*When L goes over the critical length L=2.292m, the wave run-up for the case H =0.04 m decreases with the increase of the gentle slope length, but results of the case H = 0.06 m still increase with the increase of the gentle slope length. Moreover, the main notable result is that the normal cliff gives an increase result and the toe-eroded cliff gives an opposite result for cases H = 0.05 m.*"

    were changed to

    "*When L goes over the critical length 2.292m, the wave run-up fluctuates. When L > 2.292 m, for both normal cliff and toe-eroded cliff, run-up of the case H = 0.04 m decreases, but results of the case H = 0.06 m still increase with the increase of the gentle slope length. Moreover, for cases H = 0.05 m, the normal cliff gives a result of increase and the toe-eroded cliff gives an opposite result.*"

92. **Page 15, Section 4 line 22**

The sentence

"*The increase of the incident wave height and the erosion at the cliff toe exaggerates the critical value of submarine gentle slope length.*"

was deleted.

93. **Page 15, Section 4 line 22**

The sentence

"*As a whole, the run-up on a normal cliff is higher than that on a toe-eroded cliff.*"

was changed to

"*On the other hand, the run-up on normal cliff is higher than on toe-eroded cliff.*"

94. **Page 15, Section 4 line 23**

The words "*possible reason*" was changed to "*reason*".

95. **Page 15, Section 4 line 24-26**

The sentences

"*Wave can easily climb up on the normal cliff and regurgitate slowly along the cliff. As for the toe-eroded cliff, wave reflects on the cliff and only part of water can run up on the cliff, finally, under the action of gravity, water falls back earlier.*"

were inserted.

96. **Page 17, Section 4 line 6**

The sentence

"*Figs. 8(a), (c), (e) , (g), (i) and (k) give the predicted results at location P3, whereas the results at location P4 shown in Figs. 8 (b), (d), (f), (h), (j) and (l).*"

was changed to

"*Figs. 10 (a), (c), (e), (g), (i) and (k) give the predicted results at location P3, and the results at location P4 are shown in Figs. 10 (b), (d), (f), (h), (j) and (l).*"

97. **Page 17, Section 4 line 8**

The words "*wave impact*" were changed to "*directly impact*".

98. **Page 17, Section 4 line 10-11**

The sentence

"*In the action period, the main scope of relative pressure ranges from 1.5 to 2.5, taking no account of the pressure peak.*"

was changed to

"*In the action period, besides the pressure peak, the main scope of relative pressure ranges from 1.0 to 2.5.*"

99. **Page 17, Section 4 line 14**

The sentence

"*The inclination of a cliff affects the appearance of the pressure peak.*"

was changed to

"*It can be found that the inclination of cliff affects the appearance of the pressure peak.*"

100. **Page 17, Section 4 line 21**

The word "*simulated*" was changed to "*shown*".

101. **Page 17, Section 4 line 21-22**

The sentence

"*The predicted results for the case 12 are shown in Figs.10 (a), (c) and (e), whereas case 24 shown in Figs. 10 (b), (d) and (f).*"

was changed to

"*The predicted results for the case 12 are shown in Figs.12 (a), (c) and (e), and case 24 are shown in Figs. 12 (b), (d) and (f).*"

**102. Page 17, Section 4 line 22**

The words "*approaches and*" were deleted.

**103. Page 17, Section 4 line 23**

The word "*near*" was changed to "*at*".

**104. Page 19, Section 5, line 10-Page20, line 9**

The following Section was inserted, corresponding following Sections' number was changed.

"*5 Discussion*

*The tsunami amplification factor is essentially a kind of relative wave height, which normalized to the height at a reference location. The interesting result in present work is as follow. In Tank 1, we analyse the tsunami amplification factor near the steepest cliff and find that it increases with the increase of initial wave height (as Figs. 6 (e) and (f) shown). However, in Tank 2, when the gauge is close to the normal cliff, the relative wave height decrease as the increase of initial wave height (seen in Figs. 8 (a), (c) and (e)). It seems that the results of Tank1 and Tank2 are contradictory. One of the explanations is the influence of the beach. The most significant difference between Tank1 and 2 is the length of beach. The effect of beach can be simply summed up as follow. The longer the beach is, the more energy lost before wave impact. The beach is also an area for the mixing of incident and reflected wave, for a large wave which requires a long area to mixing, when the beach is not long enough, the drastic mixing will occur under the coastal line. The Details of beach effect, including process of mixing and energy dissipation, is a meaningful research subject, which remains to the future work.*

*As for the run-up in Tank 2, a critical length of submarine gentle slope is found for some cases. Before the wave gets across the coastal line, submarine gentle slope facilitates wave deformation and energy focus. A proper slope helps wave to get an adequate preparation before it touch the cliff. When the slope is too long, as wave getting the shore line, it may have broken be on the verge of breaking, which makes energy dissipate ahead of time. However, the optimum length is effect by several factors such as initial wave height and cliff slope, that's why there is no critical value found in some cases. From the present work, it is reasonable to speculate that higher initial wave require longer submarine gentle slope to achieve the critical value. This can be connected with the analysis of Fig. 11 that when L is small, small wave generates extreme pressure, and when L becomes large, high wave has a trend to generate extreme pressure. On the other hand, normal cliff also enlarge the critical value comparing with toe-eroded cliff. The complicated relationship between these factors needs an even deeper investigation.*

*The present work is only a start of future work, understanding of tsunami inundation need to be more detailed and quantificational.*"

**105. Page 20, Section 6, line 15**

The words "*angle of cliff slope*" were changed to "*gradient of cliff slope*".

**106. Page 20, Section 6, line 15**

The words "*with different incident wave heights*" were deleted.

**107. Page 20, Section 6, line 19**

The words "*in front of*" were changed to "*near*".

**108. Page 20, Section 6, line 19**

The words "*large amplification factor*" were changed to "*larger relative wave height*".

**109. Page 20, Section 6, line 19**

The words "*a normal cliff*" and "*a toe-eroded cliff*" were changed to "*normal cliff*" and "*toe-eroded cliff*", respectively.

**110. Figures**

The following figures were inserted, corresponding following figures' number was changed.

[Figure]

*Fig. 1 The principle of CIP method: (a) The solid line is the initial profile and the dashed line is an exact solution after advection, (b) discretized points after advection, (c) linearly interpolated, (d) interpolated using CIP method.*

**111. Figures**

The following figure was inserted, corresponding following figures' number was changed.

[Figure]

*Fig. 2 CIP scheme as a kind of Semi-Lagrangian method*

**112. Figures**

The following figure was redrawn.

[Figure]

*Fig. 3 Schematic diagram of Tank 1. $\tan\theta_1 = 25/17$, $\tan\theta_2 = 1/15$, $\tan\theta_3 = 1/30$.*

**113. Figures**

The following figures were redrawn.

[Figure]

Fig. 4 Schematic diagram of Tank 2. $\tan\theta_1 = 1.38$, $\tan\theta_2 = 0.08$, $\tan\theta_3 = 0.02$.

**114. Figures**

The following figures were enlarged.

"

Fig. 6 Wave amplification factors, $H_m/H_r$ of different cliff: (a) x = 0.0 m, (b) x = 0.06 m, (c) x = 0.11 m, (d) x = 0.13 m, (e) x = 0.16 m, (f) x = 0.21 m"

**115. Figures**

The following figures were enlarged.

[Figure]

*Fig. 8 Maximum relative wave height in front of the cliff in Tank 2: (a) and (b) H = 0.04 m, (c) and (d) H = 0.05 m, (e) and (f) H = 0.06 m.*"

”

**Response to the reviewers**

The authors are deeply grateful to the two reviewers for their helpful suggestions and comments. In the following part we answer on the general and specific comments.

COMMENTS TO THE AUTHOR:

**Reviewer #1**: *The paper present a numerical study of the run up on cliffs located at the back of a beach. In order to do the experiments the authors propose the use of an "in-house" model based on VOF techniques and a numerical channel of 1 m long. In order to test the validity of the model, authors reproduce experiments published by Sim et al., 2015. In these experiments, solitary waves of different heights are propagated over four segments profiles (sea floor, continental slope, continental shelf, beach, and cliff) obtaining results accordingly to the previous observations. Next, Authors investigate the effect of cliff slope on the run up using the same channel configuration and varying the cliff slope. On a second channel setup, Authors investigate the effect of continental shelf gentle slope with four different solitary waves and two cliff slopes. Finally, some results are explained.*
*Although model results are worthy and validation with laboratory experiments are also good in my opinion the paper is not suitable for publication as it is for the next reasons:*

**Q: a)** *The paper applies an 'in-house' CIP model, probably a VOF. Although this is not completely clear, model description is very poor. For example is not clear what are the differences and advantages of using this model instead of a traditional VOF.*
**R: a)** The suggestion is valuably. We need to explain the model detail more clearly, and it has been done in section 2 according to the reviewer's suggestions. CIP is a compact high order difference scheme used in the model to solve the advection term. We also employed a VOF-type method, THINC/SW, to capture the free surface. Description and advantage of CIP and THINC/SW are added in subsections 2.3 and 2.4, respectively. Some references about CIP are also introduced in Introduction, Page 2, Lines 21-31.

**Q: b)** *Regarding the experiments setup it is not stated the scale of the channel, moreover being numerical experiments, why experiments should be scaled? Considering the water depth of the channel (0.35 m) and a complete ocean profile (from the coast to deep ocean) a scale of the order of 1:10000 can be guessed. This lead to continental shelf length of 15 km and depth of 1 km (typical values for these are 80 km and 200 m). The use of these values must be justified. The wave heights used on simulations are one order of magnitude less than the ocean depth i.e., total depth h=0.35 and tsunamis waves H from 0.025 to 0.06 (For example a wave of 0.025 scaled to 1:10000 gives a wave of 250 m). This is completely unrealistic for tsunamis. In the case of laboratory experiments, some concessions must be done, but on numerical experiments there is no reason to do this. If there is a numerical or mathematical reason to scale the waves like this, it must be stated.*
**R: b)** The scale is important in both laboratory experiment and CFD. Limited by the computational efficiency and numerical dissipation of CFD, the quantity and slenderness ratio of computational grids should be moderate. Because of the high proportion of spatial span in horizontal and vertical in most cases, reasonable abnormal model scale in horizontal and vertical

is necessary, similar to laboratorial experiment. This is added in Introduction, Page 3, Lines 21-25. And the selection of scale is same with the laboratory experiment done by Sim and Huang (2013, 2015, 2017), which is added in subsection 3.1, Page 8, Lines 11-12.

**Q: c)** *On section 3.4 where time evolution of wave is analyzed on tank #2, Authors find, as expected, that the longer the gentle slope is, the later the wave arrives. Since we are taking about long waves this is an obvious results and there is nothing new on the description.*
**R: c)** This obvious result with nothing new is deleted in the revision.

**Q: d)** *On section 3.5, authors evaluate the relative wave height in front of the cliff on tank #2 at different distances. Data describes a linear trend, nevertheless authors do not mention the trend values, and do not intent to find an explanation. Furthermore, was possible to find a relationship between this trend and profile slope. None of these is done by authors.*
**R: d)** Firstly, the trend values and explanation are added in subsection 4.4, Page 15, Lines 3-12. And they are also shown as follow:
"*In Figs. 8 (a) and (b), the maximum relative wave heights are greater than 2.5, and the gradients of trend lines are 2.22 and 2.16, respectively. In Figs. 8 (c) and (d), the maximum relative wave height is 2.4, the trend lines have gradients of 1.77 and 1.75, not as steep as those in Figs. 8 (a) and (b). In Figs. 8 (e) and (f), the trend lines are gentle, with gradients of 1.46 and 1.13. In summary, in the case of smaller wave height, the development of wave height along with the decrease of distance to the cliff is more obvious, and finally a larger relative water height appears. As for lager wave, rate of wave height increase is very small, especially when the cliff is toe-eroded. The possible cause of this interesting phenomenon can be explained as follow. According to the result of Fig. 7, the crest of wave elevation is produced by the mixing of incident and reflected wave. Under the condition of large wave, the reflected wave is very strong, which makes the mixing occupy a wide area on the beach. As a result, energy distribution of large wave is not as concentrated as small wave be. The energy concentration helps the small wave to produce a higher relative wave height near the cliff.*"
Secondly, we speculate that the trend may effected by the beach in front of cliff, which has been discussed in section 5, Page 19, Line 11-20. The Details remains to the future work.

**Q: e)** *On section 3.6, authors evaluate the run up on tank #2. They found that for larger continental platforms the run up is larger. This implies that (section 3.4) longer continental slopes, produce waves traveling slower and with higher run up. Authors do not discuss this idea or any other.*
**R: e)** Result is not "for larger continental platforms the run up is larger". The result is that there is a critical length of continental platforms, which produces the largest run-up. In our studies, the critical length is 2.292 m for part of cases. And for other cases, we didn't find the critical length, which may be caused by the different wave height and cliff inclination. This has been explained more clearly in subsection 4.5, Page 15, Lines 15-26. Besides, discussion about these has been added in section 5, Page 19, Lines 21- Page 20, Line 7.

**Q: f)** *In conclusion, description of results is very poor, there is no discussions and conclusions are just a summary of the remarkable results.*

**R: f)** Discussion has been added in section 5, Pages 19-20.

**Q: e)** *Finally it is rather hard going and the English needs to be improved considerably.*
**R: e)** The English expression of the whole text has been improved in the revision.

**Reviewer #2:** *This paper has presented some interesting results of tsunami wave run-up and impact on the coastal cliffs, which provide a valuable contribution to prevent the damage caused by tsunami. The content is clear, concise and well-presented. However, the text needs to be significantly improved before consideration for publication and these comments are outlined below.*

**Q: 1)** *Please provide a more detailed description of the numerical CIP method and THINC/SW scheme in section 2.*
**R: 1)** Description and advantage of CIP and THINC/SW are added in subsections 2.3 and 2.4, respectively. Some references about CIP are also introduced in Introduction, Page 2, Lines 21-31.

**Q: 2)** *In section 3.2, the authors should describe the method they used to define and create the solitary wave.*
**R: 2)** The subsection "*3.2 Numerical wave-maker*" has been added from Page 9, Line 1 to Page 10, Line 3.

**Q: 3)** *Please ensure that all figures appear on the top of each page.*
**R: 3)** It has been improved.

**Q: 4)** *The English expression of the whole text needs to be improved.*
**R: 4)** The whole text has been reviewed carefully and the English expression has been improved.

**Q: 5)** *Please ensure on your next submission that the line numbers appear throughout the whole document rather than showing just 5, 10, 15, 20 and 25.*
**R: 5)** The line numbers has been corrected in the revison.

**Q: 6)** *Please make sure that Fig.1 and Fig. 2 are presented clearly.*
**R: 6)** These Figures have been redrawn more clearly.

**Q: 7)** *Please enlarge the font in Figures 1, 2, 4 and 6 so that these figures are readable.*
**R: 7)** The font in Figs. 1 and 2 has been enlarged. Figs 4 and 6 have been enlarged in whole.

**Q: 8)** *In Page 5, Line 2, "The angles of slopes are $\tan\theta1=25/17$, $\tan\theta2=1/15$, $\tan\theta3=1/30$" is a wrong expression.*
**R: 8)** The sentences "*The angles of slopes are $\tan\theta_1= 25/17$, $\tan\theta_2= 1/15$, $\tan\theta_3= 1/30$.*" and "*The angles of slopes are $\tan\theta_1 = 1.38$, $\tan\theta_2 = 0.08$, $\tan\theta_3 = 0.02$.*" have been deleted. New expressions are added in the Figure captions "*Fig. 3 Schematic diagram of Tank 1. $\tan\theta_1= 25/17$, $\tan\theta_2= 1/15$ and $\tan\theta_3= 1/30$.*" and "*Fig. 4 Schematic diagram of Tank 2. $\tan\theta_1 = 1.38$, $\tan\theta_2 = 0.08$ and $\tan\theta_3 = 0.02$.*".

**Q: 9)** *Please also check the description in Page 5, Line 13 "The angles of slopes. . .".*

**R: 9)** It has been improved as shown in "**R: 8)**".

**Q: 10)** *In Page 5, Line 15, the sentence should be "Two kinds of cliff, normal cliff of $\theta4$ =80.02◦degree and toe-eroded cliff $\theta4$ = 91.91◦ are considered."*

**R: 10)** It has been improved in the revision.

**Q: 11)** *In Page 5, Line 4, "Solitary waves are" should be "Solitary wave is".*

**R: 11)** It has been corrected in the revision.

**Q: 12)** *In the whole text, the authors can consider to use the expression of "Tank 1" and "Tank 2" not "1# tank and 2#tank".*

**R: 12)** It has been improved according to the reviewer's advice.

**Q: 13)** *In Page 6, Line 18, please improve the expression of the sentence "Then, it impacts and runs upon the cliff and falls back to the beach.".*

**R: 13)** The sentence has been changed to "*Then, the water jet impacts the cliff, accompanied by large pressure acting on the toe of cliff. Great acceleration is produced by the impact, making the water run up on the cliff. Under the action of gravity, water finally falls back, large quantity of air is entrained in water when backflow interacts with the incident flow.*".

**Q: 14)** *In Page 7, Line 6, "However, for the" should be "However, as the".*

**R: 14)** It has been corrected in the revision.

**Q: 15)** *In Page 7, Line 19, "As for Fig. 4(e) and (f)" should be "As for Figs. 4(e) and (f)".*

**R: 15)** It has been corrected in the revision.

**Q: 16)** *In Page 8, Lines 3-4, the velocity of the water particle cannot be faster, it should be higher.*

**R: 16)** It has been corrected in the revision.

**Q: 17)** *In Page 9, Line 4, "Figs. 5(b), (d), (f)" should be "Figs. 5(b), (d) and (f)", "Figs. 5(a), (c), (e)" should be "Figs. 5(a), (c) and (e)".*

**R: 17)** It has been corrected in the revision.

**Q: 18)** *In Page 10, Line 2, "x=0m, 0.1m, 0.2m, 0.3m, 0.4m" should be "x=0m, 0.1m, 0.2m, 0.3m and 0.4m".*

**R: 18)** It has been corrected in the revision.

**Q: 19)** *In Page 10, Lines 4-5, there is no number after the word of "Figs".*

**R: 19)** It has been changed to "*Figs. 8*" in the revision.

**Q: 19)** *Please complete and improve the introduction and literature review with more recent journal publications.*

**R: 19)** More recent journal publications have been discussed in Introduction. And they are also shown as follow.

"*Finite difference method is widely used in various CFD models as flow solver. Hitherto, the accuracy of finite difference method is still a great challenge. In this paper, we introduce a CFD model based on constrained interpolation profile (CIP) algorithm. The CIP method was first introduced by Takewaki et al. (1985) as a high order method to solve the hyperbolic partial differential equation. Tanaka et al. (2000) proposed a new version of the CIP-CSL4, which overcomes the difficulty of conservative property. Hu et al. (2009) simulated strongly nonlinear wave-body interactions used a CIP-based Cartesian grid method, and the results were in good agreement with experiment data. Kawasaki et al. (2015) developed a tsunami run-up and inundation model based on CIP method, and high accurate water surface profile was observed by using slip condition on the wet-dry boundary. Fu et al. (2017) simulated the flow past an in-line forced oscillating square cylinder by a CIP-based model. CIP method can be not only applied in CFD, but also has good performance in other areas. Sonobe et al. (2016) employed CIP method to simulate sound propagation involving the Doppler-effect.*"

"*Yokoi et al. (2013) proposed a numerical framework consisting of CLSVOF method, multi-moment methods and density-scaled CSF model. The framework can well capture free surface flows with complex interface geometries. More recently, conventional VOF has been widely used by combining with various additional scheme. Malgarinos et al. (2015) proposed an interface sharpening scheme on the basic of standard VOF method, which effectively restrained interface numerical diffusion. Gupta et al. (2016) used a coupled VOF and pseudo transient method to solve free surface flow problems, and the numerical solution was compared well with analytical or experimental data. Quiyoom et al. (2017) simulated the process of gas-induced liquid mixing in a shallow vessel, found that the mixing time predicted by EL+VOF was in good agreement with the measurements.*"

"*Markus et al. (2014) introduced a Virtual Free Surface (VFS) model, which enabled the simulations of fully submerged structures subjected to pure waves and combined wave–current scenarios. Vicinanza et al. (2015) proposed new equations to predict the magnitude of forces exerted by the wave on its front face. The equations were added in 5 random wave CFD model and good agreement was obtained when compared with empirical predictions. Oliveira et al. (2017) utilized PFEM to simulate complex solid-fluid interaction and free surface, so that a piston numerical wave-maker are implemented in a numerical wave flumes. Regular long wave was successfully generated in the numerical wave flumes.*"

**A numerical study of tsunami wave impact and run-up on coastal cliffs using a CIP-based model**

Xizeng Zhao[1], Yong Chen[1], Zhenhua Huang[2], Zijun Hu[1], Yangyang Gao[1]

[1] Ocean College, Zhejiang University, Zhoushan Zhejiang 316021, China

[revised manuscript text omitted]

Since the available time-series data of tsunami waveform and flow field in the near-shore area is scarce, there is a general lack of understanding of tsunami interacting with complex geography. A more accurate method is required to reproduce the process of tsunami evolution in the coastal area. Due to the development of supercomputing technology and precise numerical algorithm, computational fluid dynamics (CFD) with viscous flow theory and fluid-solid coupling mechanics are capable of dealing with the complex flow problems when geographies exist. Finite difference method is widely used in various CFD models as flow solver. Hitherto, the accuracy of finite difference method is still a great challenge. In this paper, we introduce a CFD model based on constrained interpolation profile (CIP) algorithm. The CIP method was first introduced by Takewaki et al. (1985) as a high order method to solve the hyperbolic partial differential equation. Tanaka et al. (2000) proposed a new version of the CIP-CSL4, which overcomes the difficulty of conservative property. Hu et al. (2009) simulated strongly nonlinear wave-body interactions used a CIP-based Cartesian grid method, and the results were in good agreement with experiment data. Kawasaki et al. (2015) developed a tsunami run-up and inundation model based on CIP method, and high accurate water surface profile was observed by using slip condition on the wet-dry boundary. Fu et al. (2017) simulated the flow past an in-line forced oscillating square cylinder by a CIP-based model. CIP method can be not only applied in CFD, but also has good performance in other areas. Sonobe et al. (2016) employed CIP method to simulate sound propagation involving the Doppler-effect.

It is a significant research project to deal with the free surface problem in CFD. Hirt et al. (1981) put forward a mass conservation method named Volume of Fluid (VOF), which is flexible and efficient for treating complicated free boundary configurations. Based on the principle of VOF, several improved methods were developed: PLIC (Youngs, 1982), THINC

(Xiao et al., 2005), WLIC (Yokoi, 2007) and THINC/SW (Xiao et al., 2011). Yokoi et al. (2013) proposed a numerical framework consisting of CLSVOF method, multi-moment methods and density-scaled CSF model. The framework can well capture free surface flows with complex interface geometries. More recently, conventional VOF has been widely used by combining with various additional scheme. Malgarinos et al. (2015) proposed an interface sharpening scheme on the basic of standard VOF method, which effectively restrained interface numerical diffusion. Gupta et al. (2016) used a coupled VOF and pseudo transient method to solve free surface flow problems, and the numerical solution was compared well with analytical or experimental data. Quiyoom et al. (2017) simulated the process of gas-induced liquid mixing in a shallow vessel, found that the mixing time predicted by EL+VOF was in good agreement with the measurements.

When coastal geographies are included, special handling of solid boundary is required. Peskin (1973) proposed an immersed boundary method (IBM) to treat the blood flow patterns of human heart, and was later introduced to simulate the interactions between solid objects and incompressible fluid flows (Ha et al., 2014; Lin et al., 2015).

CFD is more convenient and economical than laboratorial experiment and field observation. The most attractive feature is that it can provide time-series data of waveform and flow field, which are helpful for a better understanding of the inundation mechanism of wave in near-shore areas. Markus et al. (2014) introduced a Virtual Free Surface (VFS) model, which enabled the simulations of fully submerged structures subjected to pure waves and combined wave–current scenarios. Vicinanza et al. (2015) proposed new equations to predict the magnitude of forces exerted by the wave on its front face. The equations were added in 5 random wave CFD model and good agreement was obtained when compared with empirical predictions.Oliveira et al. (2017) utilized PFEM to simulate complex solid-fluid interaction and free surface, so that a piston numerical wave-maker are implemented in a numerical wave flumes. Regular long wave was successfully generated in the numerical wave flumes.

When efficient numerical algorithm is adopted, CFD numerical simulation can be applied to study the tsunami inundation in coastal areas. However, limited by the computational efficiency and numerical dissipation of finite difference method, the quantity and slenderness ratio of computational grids should be moderate. Because of the high proportion of spatial span in horizontal and vertical in most cases, reasonable abnormal model scale in horizontal and vertical is necessary, similar to laboratorial experiment.

The purpose of present work is to understand the characteristics of a tsunami, helping inverse the generation mechanisms and provide reference for the tsunami forecasting and post-disaster treatment. In this study, tsunami wave impact and run-up on coastal cliffs is simulated using an in-house code, named the CIP-Based model. Considerable attention is paid to the influence of different coastal topographies, steep cliffs on the beach and submerged gentle slopes are considered. Coastal cliffs are one of the common coastal landforms, representing approximately 75% of the world's coastline (Rosser et al., 2005), such as the coast of Banda Aceh in Indonesia and the steep slope at San Martin (António et al., 1993). The existence of cliff can not only influence the impact and run-up of a tsunami wave but also the erosion-deposition. Different layers provide variations in resistance to erosion (Stephenson et al., 2011). Particularly, some coastal cliffs consisting of soft rocks are eroded at the toe (Yasuhara et al., 2002), which makes it easier to be destroyed. It is indispensable to understand the

1 function of coastal cliffs and submerged gentle slopes during the tsunami wave approaches near shore. The submerged gentle

2 slope such as the continental shelf, affects the waveform evolution and wave celerity before tsunami waves reach the

3 shoreline. Tsunami amplification factor (Satake, 1994), relative wave height, run-up on the cliff and impact pressure will be

4 analyzed in this work.

5 In this paper, Section 2 describes the governing equations and the numerical methods; Section 3 provides the initial

6 condition and numerical wave-maker; Section 4 presents model validation; Dimensionless analysis is then used to examine

7 the effect of front slope length, depth ratio and cliff angles on the run-up, and impact pressure; finally, the discussion and

8 conclusion are made in Section 5 and 6, respectively.

9 **2 Numerical models**

10 **2.1 Governing equations**

11 Our model is established in a two-dimension Cartesian coordinate system, based on viscous fluid theory with incompressible

12 hypothesis. The governing equations are continuity equation and Navier-Stokes equations written as follows

$$\nabla \cdot \boldsymbol{u} = 0 \tag{1}$$

$$\frac{\partial \boldsymbol{u}}{\partial t} + (\boldsymbol{u} \cdot \nabla)\boldsymbol{u} = -\frac{1}{\rho}\nabla p + \frac{\mu}{\rho}\nabla^2 \boldsymbol{u} + f \tag{2}$$

15 where $\boldsymbol{u}$, $t$, $\rho$, $p$, $\mu$ and $f$ are the velocity, time, fluid density, hydrodynamic pressure, dynamic viscosity and momentum

16 forcing components, respectively.

17 Multiphase flow theory is employed to solve the problem of solid-liquid-gas interaction. A volume function $\phi_m$ is defined

18 to describe the percentage of each phase in a mesh

$$\frac{\partial \phi_m}{\partial t} + \boldsymbol{u} \cdot \nabla \phi_m = 0 \tag{3}$$

20 where $m$ = 1, 2, 3, indicating liquid, gas, solid respectively, and $\phi_1 + \phi_2 + \phi_3 = 1$.

21 Physical property, such as the density and viscosity in a mesh, can be calculated by:

$$\lambda = \sum_{m=1}^{3} \phi_m \lambda_m \tag{4}$$

23 **2.2 The fractional step approach**

24 A fractional step approach is applied to solve the time integration of the governing equations (1) and (2). The first step is to

25 calculate the advection term, neglecting the diffusion term and pressure term, as Eq. (5) shows.

$$\frac{\partial \boldsymbol{u}}{\partial t} + (\boldsymbol{u} \cdot \nabla)\boldsymbol{u} = 0 \tag{5}$$

[Figure]

2    Fig. 1 The principle of CIP method: (a) The solid line is the initial profile and the dashed line is an exact solution after advection, (b)

3         discretized points after advection, (c) linearly interpolated, (d) interpolated using CIP method.

4    A CIP (Constrained Interpolation Profile) method is employed to solve Eq. (5). The second step is to solve the diffusion

5 term by a central difference scheme

$$\frac{\boldsymbol{u}^{**} - \boldsymbol{u}^{*}}{\Delta t} = \frac{\mu}{\rho} \nabla^2 \boldsymbol{u} + \vec{F} \tag{6}$$

7 where $\boldsymbol{u}^{*}$ is the solution of Eq. (5), and $\boldsymbol{u}^{**}$ is the solution to calculate in this step.

8    The final step is the coupling of the pressure and velocity by considering Eq. (1):

$$\nabla \cdot (\frac{1}{\rho} \nabla p^{n+1}) = \frac{1}{\Delta t} \nabla \cdot \boldsymbol{u}^{**} \tag{7}$$

$$\boldsymbol{u}^{n+1} = \boldsymbol{u}^{**} - \frac{\Delta t}{\rho} \nabla p^{n+1} \tag{8}$$

11 Eq. (7) is solved by a successive over relaxation (SOR) method. More details can be found in our previous works (Zhao et al.,

12 2016a and 2016b).

13    The free surface is captured by a tangent of hyperbola for interface capturing with slope weighting (THINC/SW) scheme,

14 which is based on the principle of VOF method. The solid boundary is treated by an immersed boundary method (IBM)

15 (Peskin et al., 1973).

16 **2.3 CIP method**

17 The basic principle of CIP is that when computing the advection of a variable $f$, both the transportation equation of $f$ and the

18 transportation equation of its spatial gradient $g = \partial f / \partial x$ are used (seen in Fig. 1).

19    To explain the step of CIP method, we take the following 1D advection equation as an example.

$$\frac{\partial f}{\partial t} + u \frac{\partial f}{\partial x} = 0 \tag{9}$$

21    By differentiating Eq. (9) with respective to $x$, we have the equation about the spatial derivative as

$$\frac{\partial g}{\partial t} + u \frac{\partial g}{\partial x} = -g \frac{\partial u}{\partial x} \tag{10}$$

[Figure]

2                                    Fig. 2 CIP scheme as a kind of Semi-Lagrangian method

3    where $g = \partial f / \partial x$. For simplicity, we assume a constant advection velocity. Then Eq. (10) has the same form as Eq. (9). For

4    the case of $u > 0$, we may approximate a profile for $f^n$ inside the upwind cell $[x_{i-1}, x_i]$ as

5                    $$F_i^n(x) = a_i(x - x_i)^3 + b_i(x - x_i)^2 + c_i(x - x_i) + d_i$$                    (11)

6        Since, the spatial gradient of $f^n$ can be written as

7                    $$G_i^n(x) = a_i(x - x_i)^2 + b_i(x - x_i) + c_i$$                    (12)

8        As shown in Fig. 2 the profile at the time step n+1 is obtained by shifting the profile with $-u\triangle t$, i.e., the time evolution of

9    the function f and g can be obtained by using the following Lagrangian invariants.

10                   $$f_i^{n+1} = F_i^n(x_i - u \cdot \Delta t)$$                    (13)

11                   $$g_i^{n+1} = G_i^n(x_i - u \cdot \Delta t)$$                    (14)

12       Therefore we call the CIP scheme as a Semi-Lagrangian method.

13       The four unknown coefficients in Eq. (11) can be determined by using known quantities $f_i^n$, $f_{i-1}^n$, $g_i^n$, $g_{i-1}^n$. It is not difficult

14   to obtain

15                   $$a_i = \frac{g_i^n - g_i^n}{\Delta x^2} - \frac{2(f_i^n - f_{i-1}^n)}{\Delta x^3}, \quad c_i = g_i^n$$
     $$b_i = \frac{2g_i^n + g_i^n}{\Delta x} - \frac{3(f_i^n - f_{i-1}^n)}{\Delta x^2}, \quad d_i = f_i^n$$                    (15)

16       By introducing the spatial gradient of variable $f$, CIP can provide a third order interpolation function in a single grid.

17   Comparing to traditional upwind schemes, CIP method has not only sub-cell resolution, but also compact high order

18   structure.

19   **2.4 THINC/SW scheme**

20   The THINC/SW scheme, first put forward by Xiao et al. (2011), is used for free surface capturing of incompressible flows.

21   Some test examples has indicated that the scheme has the features we need: mass conservation, a lack of oscillation (Ji et al.

22   2013). The basic idea of THINC/SW is for the profile of a volume function $\phi$ ($0 \le \phi \le 1$), a hyperbolic tangent function with

23   adjustable parameter is used to interpolate inside an upwind cell, which is shown as follow:

24                   $$\chi_{x,i} = \frac{\alpha}{2}\{1 + \gamma \tanh[\beta(\frac{x - x_{i-1/2}}{\Delta x_i} - \delta)]\}$$                    (16)

1 where $\alpha$, $\gamma$, $\delta$, $\beta$, are parameters to be specified. $\alpha$ and $\gamma$ are used to avoid interface smearing, which are given by:

$$\alpha = \begin{cases} \bar{\phi}_{i+1} & \text{if } \bar{\phi}_{i+1} \geq \bar{\phi}_{i-1} \\ \bar{\phi}_{i-1} & \text{otherwise} \end{cases} \tag{17}$$

$$\gamma = \begin{cases} 1 & \text{if } \bar{\phi}_{i+1} \geq \bar{\phi}_{i-1} \\ -1 & \text{otherwise} \end{cases} \tag{18}$$

4 Parameter $\delta$ is used to determine the middle point of the hyperbolic tangent function, and is solved by:

$$\frac{1}{\Delta x_i} \int_{x_{i-1/2}}^{x_{i+1/2}} \chi_i(x) dx = \bar{\phi}_i^n \tag{19}$$

6 Parameter $\beta$ determines the steepness of the jump in the interpolation function varying from 0 to 1. In traditional THINC,
7 a constant $\beta = 3.5$ is usually used which may result in ruffling the interface which aligns nearly in the direction of the
8 velocity. To avoid this problem, in THINC/SW, $\beta$ is determined adaptively according to the orientation of the interface. In a
9 2D case, $\beta$ can be written as:

$$\begin{cases} \beta_x = 2.3 |n_x| + 0.01 \\ \beta_y = 2.3 |n_y| + 0.01 \end{cases} \tag{20}$$

**3 Simulation setup**

**3.1 Initial condition**

13 In this section, 2D numerical wave tanks including incident waves and geographies are introduced. Simulation cases are
14 divided into two categories according to different parameters and purposes.
15 The first simulations are completed in Tank 1, as shown in Fig. 3. This wave tank is 10.0 m in length and 1.0 m in height.
16 Four slopes compose the topography-profile, representing continental slope, continental shelf, beach and cliff, respectively.
17 The still water depth in front of the topography-profile is fixed at 0.35 m, so that the still-water shoreline is located at the
18 starting point of the beach. This point is regarded as the original point of this tank to determine other positions mentioned in
19 this paper. Tank 1 is used for verifying the accuracy of our model and investigating cliff slope gradient and incident wave
20 height, which may influence the tsunami amplification factor. Five cliff slopes are tested: $\theta_4 = 14^\circ$, $21.67^\circ$, $39.33^\circ$, $49^\circ$ and
21 $79^\circ$. And four incident wave heights are considered: $H = 0.025$ m, $0.035$ m, $0.045$ m and $0.055$ m. Solitary wave is used as an
22 analogue of tsunami in the numerical modelling. S1~3 in Fig. 3 are three gauges of water elevation, located at $x=$ -7.67 m, -
23 0.87 m and 0.11 m, respectively. Outcomes of S1~3 are used for the comparison between numerical results and experimental
24 data (Sim, 2017). Besides, six gauges of water elevation are employed to calculate the tsunami amplification factor, fixed at
25 $x = 0.0$ m, 0.06 m, 0.11 m, 0.13 m, 0.16 m and 0.21 m, respectively (not drawn in Fig. 3).

[Figure]

Fig. 3 Schematic diagram of Tank 1. $\tan\theta_1 = 25/17$, $\tan\theta_2 = 1/15$ and $\tan\theta_3 = 1/30$.

[Figure]

Fig. 4 Schematic diagram of Tank 2. $\tan\theta_1 = 1.38$, $\tan\theta_2 = 0.08$ and $\tan\theta_3 = 0.02$.

The second simulations utilize the Tank 2, similar to the Tank 1 except slight difference, as shown in Fig. 4 (a). In this tank, the still-water shoreline also lies on the starting point of the beach, which is the original point of this tank. Four submarine gentle slopes (standing for continental shelf) of different lengths are used: $L$ = 0.764 m, 1.528 m, 2.292 m and 3.056 m. Three incident wave heights are performed, $H$=0.04m, 0.05m, 0.06m. Two kinds of cliff, normal cliff of $\theta_4$ = 80.02° degree and toe-eroded cliff of $\theta_4$ = 91.91° degree are considered. Six gauges of water elevation are employed to record the waveform evolution, located at $x$= -0.87 m, 0.0 m, 0.1 m, 0.2 m, 0.3 m, 0.4 m, respectively. Five pressure sensors are arranged near the toe of the cliff, as shown in Fig. 2 (b). The scales of these two kinds of wave tanks are same to the previous works of (Huang et al., 2013; Sim et al., 2015; Sim, 2017).

In this work, considerable attention will be paid to Tank 2. It is necessary to number the simulated cases in Tank 2 to avoid confusion, as shown in Table 1.

Table 1 Summary of basic parameters calculated in Tank 2

| Case | $\theta_4$=80.02°
$H$=0.04 m | $\theta_4$=80.02°
$H$=0.05 m | $\theta_4$=80.02°
$H$=0.06 m | $\theta_4$=91.91°
$H$=0.04 m | $\theta_4$=91.91°
$H$=0.05 m | $\theta_4$=91.91°
$H$=0.06 m |
|---|---|---|---|---|---|---|
| $L$=0.764 m | 1 | 5 | 9 | 13 | 17 | 21 |
| $L$=1.528 m | 2 | 6 | 10 | 14 | 18 | 22 |
| $L$=2.292 m | 3 | 7 | 11 | 15 | 19 | 23 |
| $L$=3.056 m | 4 | 8 | 12 | 16 | 20 | 24 |

**3.2 Numerical wave-maker**

By declaring a velocity of water particle in the left-most grid and assigning it a value from laboratory wave-paddle velocity, a numerical paddle wave maker is set at the left side of wave tank (Fig. 3 and Fig. 4 (a)).

For a solitary wave, the approximate solution of wave profile near the wave paddle can be described as follow (Boussinesq, 1872):

$$\eta = H \operatorname{sech}^2[\sqrt{\frac{3H}{4h^3}}(ct - \xi)] \tag{21}$$

$$c = \sqrt{g(h+H)} \tag{22}$$

where $H$, $h$, $c$, $\xi$ are the amplitude of the solitary wave, still water depth, wave celerity and wave-paddle trajectory, respectively.

The wave-paddle velocity can be calculated as:

$$u_1(\xi,t) = \frac{d\xi}{dt} \tag{23}$$

For a long wave, the depth-averaged horizontal velocity of water particle derived from continuity equation is expressed as (Mei, 1983):

$$u_2(x,t) = \frac{c\eta(x,t)}{h + \eta(x,t)} \tag{24}$$

The horizontal water particle velocity adjacent to the paddle is equal to the wave-paddle velocity, which means that when $x = \xi$ in Eq. (24), $u_1 = u_2$.

Using Eqs. (24), (26) and (27), wave-paddle trajectory can be derived as an implicit expression:

$$\xi(t) = \sqrt{\frac{4H}{3h}}h\tanh[\sqrt{\frac{4H}{3h^3}}(ct - \xi)] \tag{25}$$

The stroke length of wave-paddle can be calculated as:

$$S = \xi(\infty) - \xi(-\infty) = \sqrt{\frac{16H}{3h}}h \tag{26}$$

In theory, period of solitary wave is infinite. In the application, it can be approximately define as follow:

$$\tanh[\sqrt{\frac{4H}{3h^3}}(c\frac{T}{2} - \frac{S}{2})] = 0.999 \tag{27}$$

$$T = \frac{2}{c}\sqrt{\frac{4h^3}{3H}}(3.8 + \frac{H}{h}) \tag{28}$$

The water particle velocity imposed in the left-most grid can be given by:

$$\bar{u}(t) = \frac{c\eta(\xi,t)}{h + \eta(\xi,t)} \quad 0 \le t \le T \tag{29}$$

In our model, the left-most grid is not moveable as laboratory wave-paddle be, modification should be provided to Eq. (29). By some numerical tests, it is finally determined as:

$$\bar{u}(t) = \frac{c[2\eta(\xi(t),t) - \eta(\xi(0),t)]}{h + \eta(\xi(0),t)} \quad 0 \le t \le T \tag{30}$$

**4 Numerical result**

**4.1 Model validation**

To verify the accuracy of our model, numerical result from one of the cases in Tank 1 is compared with available experimental data (Sim, 2017). The incident wave height and the cliff slope gradient of this case are $H$ =0.055m and $\theta_4 = 79^\circ$, respectively. A variable grid is used for the computation, in which the grid points are concentrated near the free surface and the topography. Three non-uniform grids are used to perform a grid refinement test. The grid quantity and the minimum grid size are shown in Table 2.

Table 2 Parameter of three sets of grids (Unit: m)

|  | Horizontal grid quantity | Vertical grid quantity | Horizontal minimum grid | Vertical minimum grid |
|---|---|---|---|---|
| Coarse-grid | 826 | 220 | 0.008 | 0.0022 |
| Middle-grid | 970 | 320 | 0.005 | 0.0015 |
| Fine-grid | 1228 | 468 | 0.003 | 0.0008 |

Fig. 5 concerns the predicted time series of water elevations at different locations S1~3 and the physical measurements (Sim, 2017) are also presented for comparison. Fig. 5 (a) illustrates the comparison results at S1. It can be observed that wave has not reached the topography, so that the waveform has not transformed grossly and is similar to the original waveform. Good general agreement is found for all computations. The relative wave height at S1 is 1.0, which reveals the accuracy of the target incident wave. Fig. 5 (b) shows the results at S2 $x$ = -0.87 m. This gauge point is in the area of the submarine gentle slope, and shoaling happens when wave propagates here. It can be seen from Fig. 5 (b) that the wave front face becomes steep and the back face becomes gentle, which means wave asymmetry appears. Results of three grids are in good agreement with experimental data. Fig. 5 (c) is the most significant among these three graphs, for the gauge station of this graph is located in the front of the cliff where the flow field is extremely complex. Wave rushes from the coastline in a shape of water jet. Then, the water jet impacts the cliff, accompanied by large pressure acting on the toe of cliff. Great acceleration is produced by the impact, making the water run up on the cliff. Under the action of gravity, water finally falls back, large quantity of air is entrained in water when backflow interacts with the incident flow. The velocities of water particles fluctuate violently due to the water-cliff interaction and the drastic water-air mixing. Intense spray of water makes it hard to measure the water elevation with a wave gauge. Hence, Sim (2017) employed three HD Pro c910 web cameras to observe

[Figure]

Fig. 5 Time series of experimental data and predicted water elevations using different grids: (a) S1, (b) S2, (c) S3.

wave transformation besides the Ultralab sensors. Data of video recordings from Sim (2017) are also presented in Fig. 5 (c) marked by ×. It can be seen from Fig. 5 (c) that the crest value of video data is 18% smaller than the value of sensor data, which reveals that it is hard to determine the true trace of water surface in such a complex condition. The result of fine-grid is between the result of video data and sensor data, the result of middle-mesh is similar to the video data, and the result of coarse-mesh is 3% smaller than video data. In general, our model shows a good performance in this verifiable example, even when the flow regime is extremely unstable. More verification of our model can been found in Zhao et al. (2014). However, as the coarse-grid has a little underestimation and the fine-grid has low time efficiency, the middle-grid will be adopted to complete the remaining case studies.

**4.2 The tsunami amplification factor in Tank 1**

Fig. 6 describes the results of the tsunami wave amplification factor in Tank 1. Tsunami amplification factor is defines as a ratio of the local tsunami height to the tsunami height at a reference location. The vertical coordinates are $H_m/H_r$, in which $H_m$ means the local wave height and $H_r$ means the reference wave height at a reference location $x_r$. The reference location in present study is $x_r$= -0.87 m (same as Sim, 2017), and the reference wave height $H_r$ is provided by present numerical result. From the results in Fig. 6, it is observed that there is a critical cliff slope about $\theta_4$=45°. When cliff slope is gentler than the critical value, the tsunami wave amplification factor increases with the increase of the cliff slope gradient. When the slope is steeper than the critical slope, the effect of the cliff slope gradient becomes insignificant. This result is similar to Sim (2017). As for the influence of incident wave height, it can be found in Figs. 6 (e) and (f), of which the wave gauges are close to the cliff. When the cliff slope is gentle, close to $\theta_4$=22°, the tsunami amplification factor increases with the decrease of the

[Figure]

Fig. 6 Wave amplification factors, $H_m/H_r$ of different cliff: (a) $x = 0.0$ m, (b) $x = 0.06$ m, (c) $x = 0.11$ m, (d) $x = 0.13$ m, (e) $x = 0.16$ m, (f) $x = 0.21$ m

incident wave height. As the cliff slope becomes steeper, the effect of different incident waves become negligible at first, and then become important. The critical cliff slope is about $\theta_4=45^o$. It is noteworthy that under the condition of steep cliff, the tsunami amplification factor increases with the increase of the incident wave height, in contrast to the gentle cliff. A possible reason for this contrary phenomenon is that the velocity of water particle in the high wave is higher, which allows the high wave easier to run up on the cliff. So that when the cliff slope is gentle, the water of high wave rushes along the cliff and reaches a rearward area, but water of small wave accumulates in the front of cliff. Then, as the cliff slopes get steep, the

[Figure]

Fig. 7 Time series of relative wave elevation in Tank 2: (a), (c) and (e) $x = 0$ m; (b), (d) and (f) $x = 0.4$ m.

so-called rearward area becomes hard to reach for the high wave. This change makes the high wave to accumulate water in the area of these two gauge stations. Moreover, the tsunami amplification factor at these two stations keeps increasing with the increase of the cliff slope angle for a given incident wave, no matter the cliff is steeper or gentler than 45°. Hence, the presence of a cliff does amplify the water elevations on the beach. The influence is particularly evident for the high wave. In present study, the largest tsunami amplification factor is 2.86, as shown in Fig. 6 (f). It is similar with the result of Sim (2017), which has a value of 2.8.

**4.3 Time evolution of relative wave elevation in Tank 2**

Fig. 7 depicts the time series of relative wave elevation in Tank 2 for the cases 1-12. Four submarine gentle slope lengths and three incident wave heights are considered. The predicted results at $x = 0$ m are shown in Figs. 7 (a), (c) and (e), whereas at $x = 0.4$ m shown in Figs. 7 (b), (d) and (f). It can be noticed in Figs. 7 (a), (c) and (e) that there are conspicuous distinctions between the incident and the reflected wave. The relative height of incident wave at $x = 0$ m increases with the decrease of the initial wave height. The reflected wave fluctuates remarkably because of the complex flow pattern, and the crest reflected wave is higher than the incident wave. As for Figs. 7 (b), (d) and (f), the wave gauges are close to the cliff, it is hard to distinguish the incident and reflected wave. The superposition of incident and reflected wave makes the crest much higher

[Figure]

Fig. 8 Maximum relative wave height in front of the cliff in Tank 2: (a) and (b) *H* = 0.04 m, (c) and (d) *H* = 0.05 m, (e) and (f) *H* = 0.06 m.

than the results of Figs. 7 (a), (c) and (e). The effect of initial wave height and length of submarine gentle slope is hard to

find from Fig. 7, which remain to the following analysis. The time series results of case13~24 are similar to case1~12, which

is not shown here.

**4.4 Relative wave height in front of the cliff in Tank 2**

Wave heights at five gauges: *x*=0m, 0.1m, 0.2m, 0.3m and 0.4m, in Tank 2 are shown in Fig. 8. The predicted wave height is

normalized to the incident wave height, *H*. The trend line is also presented as the black lines in Fig. 8. In Figs. 8 (a) and (b),

[Figure]

Fig. 9 Wave run-up on the cliff

the maximum relative wave heights are greater than 2.5, and the gradients of trend lines are 2.22 and 2.16, respectively. In Figs. 8 (c) and (d), the maximum relative wave height is 2.4, the trend lines have gradients of 1.77 and 1.75, not as steep as those in Figs. 8 (a) and (b). In Figs. 8 (e) and (f), the trend lines are gentle, with gradients of 1.46 and 1.13. In summary, in the case of smaller wave height, the development of wave height along with the decrease of distance to the cliff is more obvious, and finally a larger relative water height appears. As for lager wave, rate of wave height increase is very small, especially when the cliff is toe-eroded. The possible cause of this interesting phenomenon can be explained as follow. According to the result of Fig. 7, the crest of wave elevation is produced by the mixing of incident and reflected wave. Under the condition of large wave, the reflected wave is very strong, which makes the mixing occupy a wide area on the beach. As a result, energy distribution of large wave is not as concentrated as small wave be. The energy concentration helps the small wave to produce a higher relative wave height near the cliff. It reveals a moderate surface wave magnitude may cause enormous destruction in near-shore areas.

**4.5 Wave run-up on the cliff in Tank 2**

Fig. 9 displays the wave run-up on the cliff in Tank 2, different lengths of submarine gentle slope are compared. The predicted run-up is normalized to the incident wave height, $H$. It can be seen in Fig. 9 that there exists a critical length of submarine gentle slope about $L$=2.292m. When $L$ < 2.292 m, with a given cliff, the relative wave run-up increases as the length of gentle slope increases. When $L$ goes over the critical length 2.292m, the wave run-up fluctuates. When $L$ > 2.292 m, for both normal cliff and toe-eroded cliff, run-up of the case $H$ = 0.04 m decreases, but results of the case $H$ = 0.06 m still increase with the increase of the gentle slope length. Moreover, for cases $H$ = 0.05 m, the normal cliff gives a result of increase and the toe-eroded cliff gives an opposite result. It reveals that the critical value relates to both the incident wave height and the inclination of a cliff. On the other hand, the run-up on normal cliff is higher than on toe-eroded cliff. The maximum relative run-up on normal cliff reaches up to 4.2. The reason is that the inclination of normal cliff is accordant with the direction of the incident wave while the toe-eroded cliff is contrary. Wave can easily climb up on the normal cliff and regurgitate slowly along the cliff. As for the toe-eroded cliff, wave reflects on the cliff and only part of water can run up on the cliff, finally, under the action of gravity, water falls back earlier.

[revised manuscript text omitted]

**5 Discussion**

The tsunami amplification factor is essentially a kind of relative wave height, which normalized to the height at a reference location. The interesting result in present work is as follow. In Tank 1, we analyse the tsunami amplification factor near the steepest cliff and find that it increases with the increase of initial wave height (as Figs. 6 (e) and (f) shown). However, in Tank 2, when the gauge is close to the normal cliff, the relative wave height decrease as the increase of initial wave height (seen in Figs. 8 (a), (c) and (e)). It seems that the results of Tank1 and Tank2 are contradictory. One of the possible explanations is the influence of the beach. The most significant difference between Tank1 and 2 is the length of beach. The effect of beach can be simply summed up as follow. The longer the beach is, the more energy lost before wave impact. The beach is also an area for the mixing of incident and reflected wave, for a large wave which requires a long area to mixing, when the beach is not long enough, the drastic mixing will occur under the coastal line. The Details of beach effect, including process of mixing and energy dissipation, is a meaningful research subject, which remains to the future work.

As for the run-up in Tank 2, a critical length of submarine gentle slope is found for some cases. Before the wave gets across the coastal line, submarine gentle slope facilitates wave deformation and energy focus. A proper slope helps wave to

get an adequate preparation before it touch the cliff. When the slope is too long, as wave getting the shore line, it may have broken be on the verge of breaking, which makes energy dissipate ahead of time. However, the optimum length is effect by several factors such as initial wave height and cliff slope, that's why there is no critical value found in some cases. From the present work, it is reasonable to speculate that higher initial wave require longer submarine gentle slope to achieve the critical value. This can be connected with the analysis of Fig. 11 that when $L$ is small, small wave generates extreme pressure, and when $L$ becomes large, high wave has a trend to generate extreme pressure. On the other hand, normal cliff also enlarge the critical value comparing with toe-eroded cliff. The complicated relationship between these factors needs an even deeper investigation.

The present work is only a start of future work, understanding of tsunami inundation need to be more detailed and quantificational.

**6 Conclusions**

In this study, tsunami wave impact and run-up in the presence of submarine gentle slopes and a coast cliff are investigated numerically using a CIP-based model. Numerical results are firstly compared with available experimental data and the good agreement revealed the ability of our model to solve the complex flow field, such as wave breaking, water-air mixing and violent impact. The results can be summarized as follows.

(1) The gradient of cliff slope has a critical value about 45°, different characteristics of tsunami amplification factor has been found when the angle is greater or smaller than 45°.

(2) The length of submarine gentle slope influences the tsunami wave run-up, and has a critical value about $L = 2.292$ m in this study for some cases.

(3) When wave transforms near the cliff, the cases with small incident wave height has a larger relative wave height, which means a devastating tsunami may be caused by a moderate source.

(4) It is easier for tsunami waves to run up on normal cliff than on toe-eroded cliff.

(5) There are two opportunities for the appearance of pressure peak during the process of tsunami wave run-up and impact. One is the direct impacting pressure when tsunami waves first hit the coastal cliff, and the other is caused by the backflow from the cliff after run-up with a widely affecting area.

The present study gives time history of tsunami evolution from open sea to coastal area, which is rare in field study. Several topographies and different incident waves has been considered. Comparing with the SWE result, which may underrate and need amendment, present results can simulate the tsunami in near shore areas more accurately. The present model is helpful for tsunami forecast, dangerous prediction and post-disaster analysis. Furthermore, combining with geology knowledge, the earthquake source magnitude and generation location can be determined.

**Acknowlegements**

This work was financially supported by the National Natural Science Foundation of China (Grant Nos. 51479175, 51679212), Zhejiang Provincial Natural Science Foundation of China (Grant No. LR16E090002).